# Structural analyses of the PKA RIIβ holoenzyme containing the oncogenic DnaJB1-PKAc fusion protein reveal protomer asymmetry and fusion-induced allosteric perturbations in fibrolamellar hepatocellular carcinoma

Tsan-Wen Lu[1], Phillip C. Aoto[2], Jui-Hung Weng[2], Cole Nielsen[2], Jennifer N. Cash[3], James Hall[2], Ping Zhang[2¤], Sanford M. Simon[4], Michael A. Cianfrocco[3]*, Susan S. Taylor[1,2]*

1 Department of Chemistry and Biochemistry, University of California, San Diego, La Jolla, California, United States of America, 2 Department of Pharmacology, University of California, San Diego, La Jolla, California, United States of America, 3 Life Sciences Institute, Department of Biological Chemistry, University of Michigan, Ann Arbor, Michigan, United States of America, 4 Laboratory of Cellular Biophysics, The Rockefeller University, New York, New York, United States of America

¤ Current address: National Cancer Institute, National Institute of Health, Frederick, Maryland, United States of America
* mcianfro@umich.edu (MAC); staylor@ucsd.edu (SST)

## Abstract

When the J-domain of the heat shock protein DnaJB1 is fused to the catalytic (C) subunit of cAMP-dependent protein kinase (PKA), replacing exon 1, this fusion protein, J-C subunit (J-C), becomes the driver of fibrolamellar hepatocellular carcinoma (FL-HCC). Here, we use cryo-electron microscopy (cryo-EM) to characterize J-C bound to RIIβ, the major PKA regulatory (R) subunit in liver, thus reporting the first cryo-EM structure of any PKA holoenzyme. We report several differences in both structure and dynamics that could not be captured by the conventional crystallography approaches used to obtain prior structures. Most striking is the asymmetry caused by the absence of the second cyclic nucleotide binding (CNB) domain and the J-domain in one of the RIIβ:J-C protomers. Using molecular dynamics (MD) simulations, we discovered that this asymmetry is already present in the wild-type (WT) RIIβ$_2$C$_2$ but had been masked in the previous crystal structure. This asymmetry may link to the intrinsic allosteric regulation of all PKA holoenzymes and could also explain why most disease mutations in PKA regulatory subunits are dominant negative. The cryo-EM structure, combined with small-angle X-ray scattering (SAXS), also allowed us to predict the general position of the Dimerization/Docking (D/D) domain, which is essential for localization and interacting with membrane-anchored A-Kinase-Anchoring Proteins (AKAPs). This position provides a multivalent mechanism for interaction of the RIIβ holoenzyme with membranes and would be perturbed in the oncogenic fusion protein. The J-domain also alters several biochemical properties of the RIIβ holoenzyme: It is easier to activate with cAMP,

**Data Availability Statement:** All structure files are available from the EMDB-PDB database (accession numbers 6WJF and 6WJG).

**Funding:** The SAXS data collection was conducted at the Advanced Light Source (ALS), a national user facility operated by Lawrence Berkeley National Laboratory on behalf of the Department of Energy, Office of Basic Energy Sciences, also a DOE Office of Science User Facility under contract no. DE-AC02-05CH11231, through the Integrated Diffraction Analysis Technologies (IDAT) program, supported by DOE Office of Biological and Environmental Research. Additional support comes from the National Institute of Health project ALS-ENABLE (P30 GM124169) and a High-End Instrumentation Grant S10OD018483. The computational work used the Extreme Science and Engineering Discovery Environment (XSEDE) SDSC at the Comet GPU through allocation TG-MCB170143. This work was supported by Taiwan MOE scholarship (T.-W.L.), Ruth L. Kirschstein National Research Service Award NIH/NCI T32 CA009523 (P.C.A.), and NIH grants R01 GM34921 and R35 GM130389 (S.S.T.). The cryo-EM structural analysis reported in this publication was supported by the NIH under award number S10OD020011. The funders had no role in study design, data collection and analysis, decision to publish, or preparation of the manuscript.

**Competing interests:** The authors have declared that no competing interests exist.

**Abbreviations:** AKAP, A-Kinase-Anchoring Protein; CNB, cyclic nucleotide binding; cryo-EM, cryo-electron microscopy; D/D, Dimerization/Docking; FL-HCC, fibrolamellar hepatocellular carcinoma; GaMD, Gaussian accelerated MD; MD, molecular dynamics; MTM, membrane-targeting motif; NMR, nuclear magnetic resonance; PBC, phosphate-binding cassette; PKA, cAMP-dependent protein kinase; RMSF, root-mean-square fluctuation; SAXS, small-angle X-ray scattering; WT, wild-type.

and the cooperativity is reduced. These results provide new insights into how the finely tuned allosteric PKA signaling network is disrupted by the oncogenic J-C subunit, ultimately leading to the development of FL-HCC.

## Introduction

Fibrolamellar hepatocellular carcinoma (FL-HCC) is a rare liver cancer that commonly occurs in young adults with no chronic liver disease history. Recent studies identified a unique kinase mutation, DnaJB1-PKAc (J-C), in most of patients. This mutation due to an approximately 400–kilobase pair deletion on chromosome 19 forms a chimeric transcript of the *DNAJB1* exon 1 fused with the *PRKACA* exons 2 to 10 [1]. The mutant is translated as a stable protein where the J-domain (residues 1 to 69) of DnaJB1 is fused to the cAMP-dependent protein kinase (PKA) catalytic (Cα or C) subunit (PKAc) (residues 15 to 336) (Fig 1A). This fusion protein (J-C subunit) is uniquely expressed in tumor tissues, but not in adjacent normal tissues in FL-HCC patients. Moreover, mouse models, generated using CRISPR-Cas9 gene modification, confirmed that this chimeric protein, DnaJB1-PKAc, is oncogenic [2,3]. Structural studies of the J-C subunit (J-C) showed that its kinase core structure is nearly identical to the wild-type (WT) PKAc but has 4 extra helices at the N-terminus adjacent to the A-helix, and both proteins have similar biochemical properties [4].

PKA plays as a central role in cAMP-dependent signaling pathways, which regulate numerous biological processes in mammalian cells [5]. The activity of PKA in cells is tightly regulated by regulatory (R) subunits [5,6]. In its resting physiological state, PKA exists as an inactive holoenzyme, which is composed of 2 C subunits and 1 R subunit dimer. The activity of PKA can be stimulated by the elevated concentration of cAMP, through binding of cAMP to R subunits which unleashes the activity. Each of 4 functionally nonredundant R subunit isoforms (RIα, RIβ, RIIα, and RIIβ) has distinct quaternary holoenzyme structures, cAMP sensitivity, and cellular localization [7–11]. All 4 R subunit isoforms share a similar domain organization which consists of an N-terminal Dimerization/Docking (D/D) domain followed by an intrinsically disordered linker that connects to 2 tandem cyclic nucleotide binding (CNB) domains. Embedded within each linker is an inhibitor motif that resembles a peptide substrate and docks into the active site cleft of the C subunit in the holoenzyme. RI subunits have a pseudo-substrate inhibitor motif, while in RII subunits, this motif is a substrate that becomes phosphorylated. In each CNB domain, there is a conserved phosphate-binding cassette (PBC) that directly binds cAMP. The PBC is the signature motif of the CNB domain and is critical for cAMP activation.

Molecular dynamics (MD) simulations and nuclear magnetic resonance (NMR) data of the chimeric J-C subunit and the holoenzyme formed with RIα revealed the unique dynamic and flexible features of the J-domain [11,12]. Transcriptome research has also shown that the gene expression and protein levels of the R subunits are affected in FL-HCC [13]. In normal liver tissue, RIIβ is the predominate R subunit [14]. In tumors, however, RIα mRNA and protein levels are up-regulated, while RIIβ mRNA and protein levels are decreased compared to normal liver tissue [13,15].

To provide mechanistic insight into the functional consequences of the DnaJB1-PKAc fusion protein that might allow us to better understand how it is a driver of FL-HCC, we used single-particle cryo-electron microscopy (cryo-EM) to determine a structure of the RIIβ holoenzyme formed with J-C. This is the first cryo-EM structure of any PKA holoenzyme and also

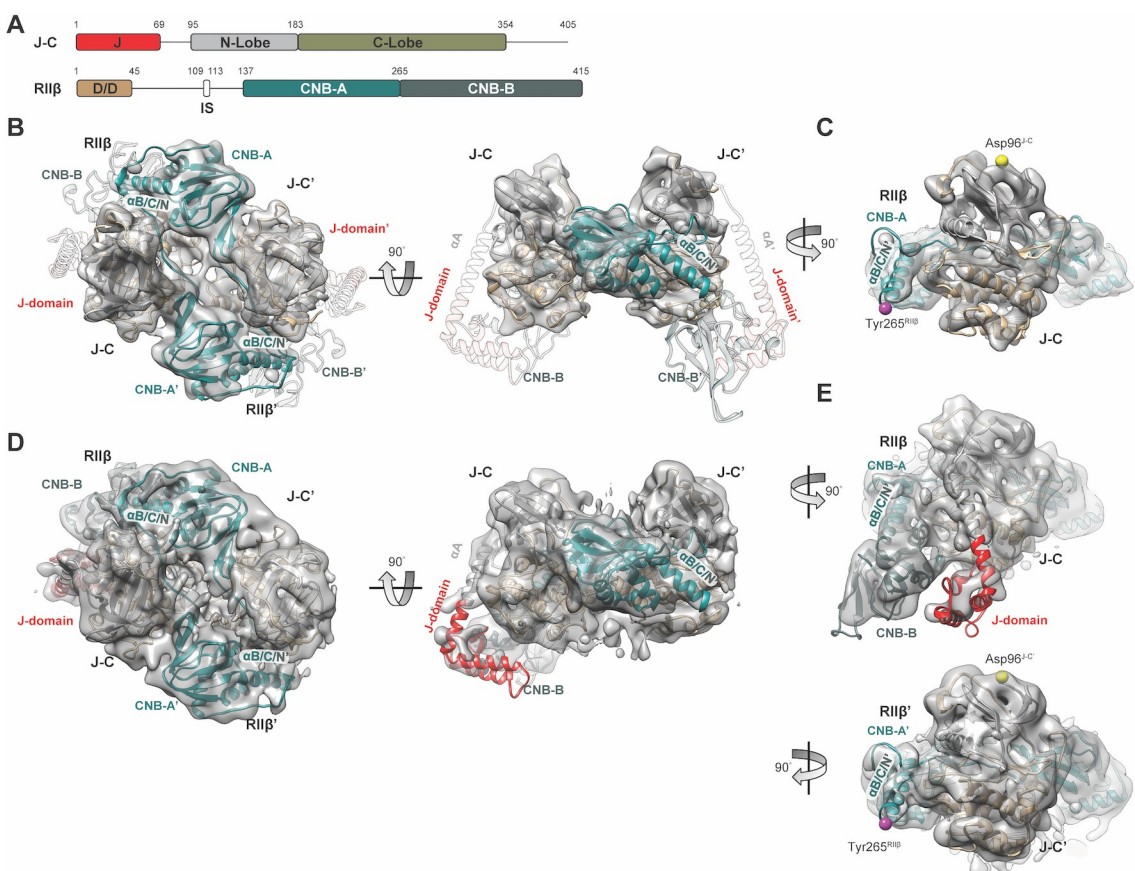

**Fig 1. Cryo-EM structure of the RIIβ₂J-C₂ holoenzyme.** (**A**) Domain diagram and color coding of RIIβ₂J-C₂ holoenzyme. (**B, C**) Cryo-EM structure the RIIβ₂J-C₂ holoenzyme (PDB = 6WJG) at 6Å with C2 symmetry imposed. (C) The EM density of the RIIβ₂J-C₂ holoenzyme starts at Asp96$^{J-C}$ (equivalent to Asp41$^C$ in the WT C subunit) in the N-terminus of the J-C subunits and ends at Tyr265$^{RIIβ}$ in the RIIβ subunit. The density for the A-helix, J-domain, and CNB-B domains are missing. (**D, E**) Structure of the RIIβ₂J-C₂ holoenzyme structure (PDB = 6WJF) after classification reveals presence of ordered J-domain and CNB-B domain. (E) One of the protomers of RIIβ₂J-C₂ holoenzyme has ordered J-domain and CNB-B domain, whereas J-domain, A-helix, and CNB-B domain are flexible in the other protomer. CNB, cyclic nucleotide binding; cryo-EM, cryo-electron microscopy; PDB, Protein Data Bank; WT, wild-type.

confirms the differences between the compact RIIβ holoenzyme and the more extended RIIα holoenzyme that was shown by negative staining EM [16]. The organization of the DnaJB1-P-KAc RIIβ holoenzyme (RIIβ₂J-C₂) is similar to what was found earlier in the previously solved crystal structure of the WT RIIβ holoenzyme (RIIβ₂C₂) [8]; the compact overall quaternary organization was not altered by the addition of the J-domain, which is also resolved in our structure. In contrast to the previous WT RIIβ holoenzyme crystal structure, where the D/D domain was not visible, the general position of the D/D domain is seen in the cryo-EM structure and confirmed by small-angle X-ray scattering (SAXS). Both our MD simulation data and the cryo-EM structure showed that the presence of the J-domain in this protein complex can change the dynamic features of holoenzyme compared to the WT RIIβ holoenzyme, especially the CNB-B domains, and both methods reveal an intrinsic asymmetry in the 2 RIIβ:J-C protomers. MD simulations of the WT RIIβ holoenzyme also reveal a dramatic asymmetry in the CNB-B domains that was hidden in the crystal structure. The asymmetry in the mutant holoenzyme is due to the flexibility in 1 protomer of the CNB-B domain in the RIIβ subunit and the J-domain and the A-helix in the J-C subunit. In addition, the chimeric protein holoenzyme

with RIIβ is activated more readily by cAMP than the WT RIIβ holoenzyme. Our studies of the $RIIβ_2J-C_2$ structure, dynamics, and function demonstrate the power of combining crystallography, MD simulations, and cryo-EM to elucidate the dynamic features of a holoenzyme complex and also provide us with a better understanding of FL-HCC that hopefully can shed light on new potential therapeutic strategies.

## Results

### Structural analysis of the DnaJB1-PKAc RIIβ holoenzyme

In the DnaJB1-PKAc chimera, the extra J-domain replaces exon 1 (residues 1 to 14) of WT C subunit and forms 4 helices with the terminal helix being contiguous with the A-helix (Fig 1A). In order to confirm the architecture of the complex, we first performed negative stain single-particle EM and single-particle analysis on the $RIIβ_2J-C_2$ holoenzyme (S1 Fig). Inspection of both micrographs and 2D class averages indicated that the $RIIβ_2J-C_2$ holoenzyme possessed a similar overall structure as the WT RIIβ holoenzyme forming a tetrameric complex with a C2 axis of symmetry at the central hole (S1B and S1C Fig). Comparison of reprojections of the WT RIIβ holoenzyme (PDB = 3TNP) with 2D averages further confirmed a similar architecture (S1B Fig). These data allowed us to conclude that the mutant $RIIβ_2J-C_2$ holoenzyme is similar to the WT RIIβ holoenzyme at low resolution.

To obtain higher-resolution information, we used cryo-EM to determine a structure of the $RIIβ_2J-C_2$ holoenzyme. Due to a preferred orientation on the cryo-EM grid, data were collected using a tilt angle of 40˚ (S2A and S2B Fig). After 2D classification and 3D reconstruction, we are able to get the structure to an average resolution of 6.2Å with C2 symmetry imposed (Fig 1B and 1C, S2C and S2D Fig, and Table 1). The structures revealed clear density in the core region of $RIIβ_2J-C_2$ holoenzyme, but density for the J-domain in the J-C subunits as well as for the CNB-B domains in the RIIβ subunits was missing (Fig 1). In addition, the density for the entire A-helix as well as the linker that wraps around the N-lobe that connects the A-helix to the β-strand 1 was missing. It is flexible with a break at Asp96$^{J-C}$ (equivalent to Asp41$^C$ in the WT C subunit) (Fig 1C). The density for the B/C/N-helix ends after the C-helix in the CNB-A domain (Tyr265$^{RIIβ}$), while density for the N-helix in the CNB-B domain is missing (Fig 1C). This is the precise break between the 2 CNB domains. Presumably, these unresolved or flexible domains are dynamic regions that form continuous states under cryo-EM conditions.

We then further masked the core regions and focused on local areas for refinement. After continued classification and refinement (S3A Fig), we were able to visualize 1 side of the J-domain and CNB-B domain together with clear density for the B/C/N-helix (Fig 1D and 1E and S3A and S3B Fig). The overall structure reveals distinct density for 1 complete RIIβ:J-C protomer, especially the contiguous B/C/N-helix (B-helix, C-helix, and N-helix), the CNB-B domain, and the J-domain (Fig 1E, top). In this protomer, the J-domain and CNB-B domain are in close proximity (Fig 1E). In contrast, the J-domain and CNB-B domain as well as the A-helix in the J-C subunit remain unresolved in the other protomer (Fig 1E, bottom). This asymmetry could be an essential mechanistic feature of the RIIβ holoenzyme. This phenomenon has not been observed previously and most likely cannot be trapped by conventional crystallography. For example, in our previous crystal structure of the RIIβ holoenzyme, the temperature factors are high in the A-helix of the C subunits and in the CNB-B domains of RIIβ subunit (S4 Fig); however, any potential asymmetry is obscured, most likely by crystal packing [8]. By using cryo-EM 3D reconstruction, we are able to observe this state that could represent an important event in PKA activation.

**Table 1. Cryo-EM data collection, refinement, and validation statistics.**

| Structure: RIIβ₂J-C₂ | | |
|---|---|---|
| **Data collection** | | |
| Grids | Gold UltrAuFoil 1.2/1.3 | |
| Vitrification method | FEI Vitrobot | |
| Microscope | Titan Krios | |
| Magnification | 29,000× | |
| Voltage (kV) | 300 | |
| Stage tilt (˚) | 40 | |
| Detector | K2 Summit | |
| Recording mode | Counting | |
| Dose rate ($e^-$/pix/sec) | 7.789 | |
| Total electron exposure ($e^-$/Å$^2$) | 77.9 | |
| Number of frames | 100 | |
| Defocus range (μm) | 1–3 | |
| Pixel size (Å) | 1.0 | |
| Number of micrographs | 1,129 | |
| Initial particle images (no.) | 642,843 | |
| **Data processing: C2 symmetry** | | |
| Final particle images (no.) | 69,605 | |
| Symmetry | C2 | |
| Map resolution (Å) | 6.2 | |
| **Data processing: C1 symmetry** | | |
| Final particle images (no.) | 11,182 | |
| Symmetry | C1 | |
| Map resolution (Å) | 7.5 | |
| **Refinement** | | |
| Initial model used (PDB code) | 3TNP, 4WB7 | |
| Symmetry | C1 | C2 |
| PDB code | 6WJF | 6WJG |
| EMDB code | EMD-21692 | EMD-23693 |
| Model resolution (Å)<br> FSC threshold | 7.5<br>0.143 | 6.2<br>0.143 |
| Map sharpening $B$ factor (Å$^2$) | −500 | −302 |
| Model composition<br> Non-hydrogen atoms<br> Protein residues<br> Ligands | <br>9,527<br>1,172<br>0 | <br>7,708<br>944<br>0 |
| $B$ factors (Å$^2$)<br> Protein (min/max/mean)<br> Ligand | <br>30/850/350<br>N/A | <br>30/550/300<br>N/A |
| RMS deviations<br> Bond lengths (Å)<br> Bond angles (˚) | <br>0.004<br>0.878 | <br>0.003<br>0.875 |
| Validation<br> MolProbity score<br> Clash score<br> Rotamer outliers (%) | <br>1.88<br>4.64<br>0 | <br>1.81<br>3.72<br>0 |
| Ramachandran plot<br> Favored (%)<br> Allowed (%)<br> Disallowed (%) | <br>85.65<br>14.35<br>0 | <br>85.36<br>14.64<br>0 |

cryo-EM, cryo-electron microscopy; EMDB, The Electron Microscopy Data Bank; FSC, fourier shell correlation; PDB, Protein Data Bank; RMS, root-mean-square.

## Molecular dynamics simulations reveal asymmetry in the RIIβ holoenzymes

To further explore the dynamic features of the 2 RIIβ holoenzymes, we used MD simulations and found surprisingly that RIIβ holoenzymes formed with WT C subunit and the J-C subunit have very different domain dynamics. Three independent 500-ns MD simulations were carried out to further explore these differences. The root-mean-square fluctuation (RMSF) analyses of each holoenzyme demonstrated distinct backbone dynamics between WT and fusion holoenzymes. The first 14 residues of the C subunit, which are typically disordered in our crystal structures and are replaced by the J-domain in the J-C fusion, are not included in these simulations; this construct is referred to as C($\Delta$1–14). The last 23 residues of the RIIβ subunit as well as the linker region and the D/D domain (residues 1 to 103) are also not included.

**Wild-type RIIβ holoenzyme.** In the simulations of the RIIβ$_2$C($\Delta$1–14)$_2$ holoenzyme, both C subunits are stable and show low RMSF throughout the entire simulation (Fig 2A) as do both CNB-A domains in the RIIβ subunit. In contrast, one of the CNB-B domains in the RIIβ subunit is highly flexible (Fig 2B). The flexible region begins approximately at residue Tyr265$^{RIIβ}$, which is at the junction of the C-helix in the CNB-A domain and the N-helix in the CNB-B domain (Fig 2C and 2D). Besides the highly dynamic features of the CNB-B domains, some local regions, such as the N-terminal linker and the β4-β5 loops in the CNB-B domains, also showed especially high RMSF. In addition, 1 small dynamic region (residues 122 to 129) was also seen in the linker that joins the inhibitor sequence to CNB-A (Fig 2B); this dynamic segment wraps around the B/C/N-helix. These regions, as well as the carboxyl terminus 23 residues, are both unresolved in the previous RIIβ$_2$C$_2$ crystal structure [8]. A deeper analysis of our simulation results further confirmed the dynamic properties of these 2 regions. By overlaying the conformational ensemble from the simulations, we can more clearly appreciate the distinct dynamic properties of the 2 CNB domains in the RIIβ$_2$C$_2$ holoenzyme (Fig 2E). One of the protomers reveals a relatively stable CNB-B domain, while the other CNB-B domain is highly flexible. In both protomers, Tyr265$^{RIIβ}$ serves as a pivot point; however, in one of the protomers, the entire CNB-B domain, including the N-helix, is extremely flexible after Tyr265$^{RIIβ}$. The MD simulation data further confirm the dynamic regions that we observed in the cryo-EM structure both being characterized by breaks at Tyr265$^{RIIβ}$.

Tyr265$^{RIIβ}$ and the dynamic properties of the B/C/N-helix are important for RIIβ holoenzyme activation. The long B/C/N-helix is the signature feature of all PKA holoenzymes. Once activated by cAMP, the B/C/N-helix divides into 3 segments (B-helix, C-helix, and N-helix) (Fig 2F and 2G). Several studies have already pointed out the importance of the flexibility of the B/C/N-helix in activation of the RIα holoenzyme [10,17,18]. Here, we show that the B/C/N-helix is also very dynamic, but different, in the RIIβ holoenzyme. Each isoform has the same hinge points, one is between the B-helix and the C-helix and the other is between the C-helix and the N-helix. Most importantly, however, the major hinge points are different (S5A Fig). The major hinge point in RIIβ, Tyr265$^{RIIβ}$, is located at the junction of the C-helix and the N-helix (Fig 2F), while Leu233$^{RIα}$, between the B- and the C-helix, is the more prominent pivot point for RIα (Fig 2G). To quantitate these differences, we measured the hinge angles of B-helix/C-helix and C-helix/N-helix in the cAMP-bound form structures of both RIα and RIIβ (PDB = 1RGS and 1CX4, respectively). The angle between the B- and C-helix is larger in RIα than in RIIβ; however, RIIβ has larger hinge angle between the C- and N-helix (Fig 2F and 2G), which is the precise junction between the 2 CNB domains.

In addition to differences in their dynamics, the B/C/N helix in RIα and RIIβ, which share 53% sequence identity, also have distinct helical, N-capping, and C-capping propensities (S5B–S5D Fig). The B/C/N-helix of RIα has very high helical propensity with a local minimum

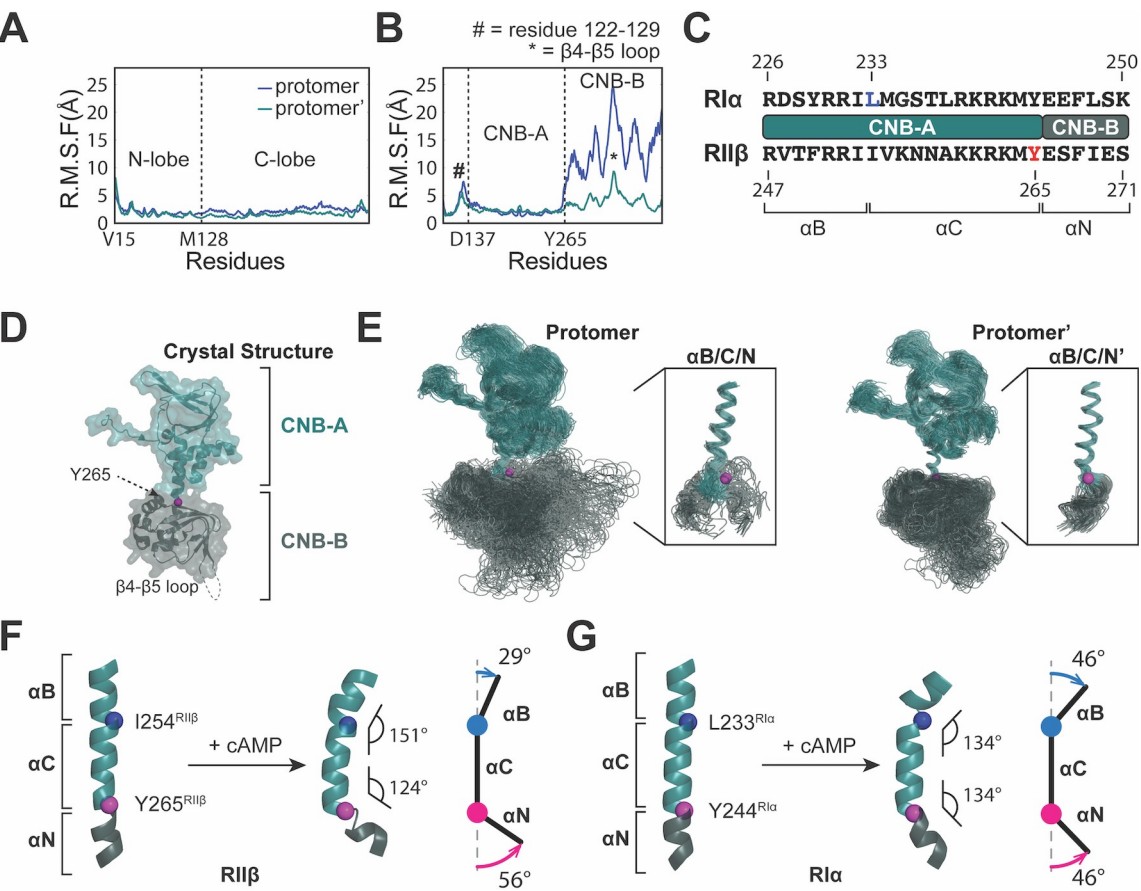

**Fig 2. MD simulations of the RIIβ₂C₂ holoenzyme reveal functional and isoform-specific dynamics.** (**A**) Both N- and C-lobe of the C subunit remain stable in the RMSF analysis of RIIβ₂C₂ holoenzyme. (**B**) CNB-B domain of the RIIβ subunit in one of the protomers is flexible in the RMSF analysis in RIIβ₂C₂ holoenzyme. (**C**) Domain diagram and residues of the B/C/N-helix in RIα and RIIβ subunit. (**D**) RIIβ subunit in the RIIβ₂C₂ holoenzyme crystal structure (PDB ID = 3TNP). (**E**) The overlaid of all states of each RIIβ subunit protomer in the RIIβ₂C₂ holoenzyme from MD simulations. CNB domain in one of the protomers is more flexible than the other, and both of the protomers have breakages at Tyr265. Residue Tyr265$^{RIIβ}$ was shown as pink ball. (**F, G**)The conformational change of RIIβ (F) and RIα (G) B/C/N-helices upon cAMP stimulation. The main pivot point of RIIβ B/C/N-helix is at Tyr265$^{RIIβ}$ (G), while the main pivot point of RIα B/C/N-helix is at Leu233$^{RIα}$ (G). The data used to make these figures can be found in S1 Data. CNB, cyclic nucleotide binding; MD, molecular dynamics; PDB, Protein Data Bank; RMSF, root-mean-square fluctuation.

at Gly235$^{RIα}$, which is near the primary hinge point (Leu233$^{RIα}$) for the cAMP-bound RIα [10]. In RIIβ, the helical propensity of the B/C/N-helix is much lower compared to RIα, perhaps allowing it to take advantage of the natural hinge point between the 2 CNB domains, Tyr265$^{RIIβ}$ (Fig 2E and S5B Fig).

**RIIβ₂J-C₂ holoenzyme.** The RIIβ₂J-C₂ holoenzyme simulations showed different dynamics compared to WT holoenzyme. Both the N- and C-lobes of J-C subunit are as stable as WT; however, the extra J-domain in the J-C subunits shows much higher RMSF values than the kinase portion of the fusion protein (Fig 3A). The addition of the J-domain in the J-C subunit also has an effect on the dynamics of the RIIβ subunit. Unlike the CNB-B domains in the RIIβ₂C₂ holoenzyme, both CNB-B domains in RIIβ₂J-C₂ holoenzyme remain strikingly more stable during the simulation (Fig 3B and S6 Fig). The J-domain presumably can stabilize the CNB-B domains either through spatial steric effects or direct interactions that are driven by the close proximity. This close communication between the J-domain and the CNB-B domain is captured in one of the protomers in our cryo-EM structure (Figs 1E and 3C). The close

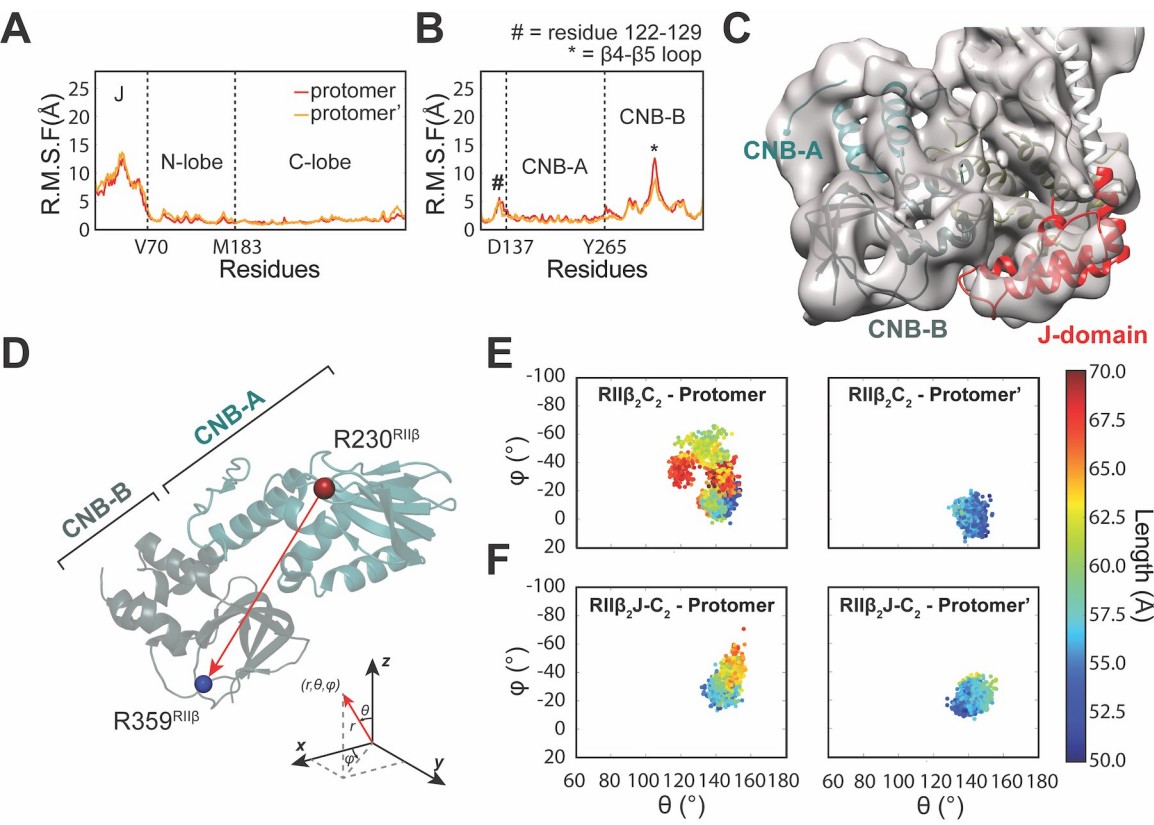

**Fig 3. MD simulations of the RIIβ₂J-C₂ holoenzyme reveal distinct dynamics.** (**A**) Both N- and C-lobe of J-C subunit remain stable but J-domain is flexible in the RMSF analysis of the RIIβ₂J-C₂ holoenzyme. (**B**) CNB-B domain of the RIIβ subunit in both protomers remain stable in the RMSF analysis in RIIβ₂C₂ holoenzyme. (**C**) J-domain and CNB-B domain of the RIIβ₂J-C₂ holoenzyme are in close proximity. (**D**) The representation of polar coordinate vector from Arg230$^{RIIβ}$ to Arg359$^{RIIβ}$. (**E, F**) The plots of φ, θ, and length of the vectors from Arg230$^{RIIβ}$ to Arg359$^{RIIβ}$ in MD simulations of RIIβ₂C₂ (E) and RIIβ₂J-C₂ holoenzymes (F). The vector in one of the protomers of the RIIβ₂C₂ holoenzyme moves and fluctuates significantly (E), whereas the vectors in both protomers of the RIIβ₂J-C₂ holoenzyme remain stable (F). The data used to make these figures can be found in S1 Data. CNB, cyclic nucleotide binding; MD, molecular dynamics; RMSF, root-mean-square fluctuation.

proximity of the J-domain and the CNB-B domain suggests that their dynamic properties are related; clearly, the CNB-B domain can sense the presence of the J-domain. The other feature that is revealed by the cryo-EM structure is that the A-helix in 1 J-C subunit is also missing, suggesting that the J-domain influences not only the CNB-B domain but also the A-helix that it is directly fused to β-strand 1 in the N-lobe (Fig 3C).

We can also visualize the differences in dynamics of the CNB domains in the DnaJB1-PKAc and WT RIIβ holoenzymes by measuring the polar vector between CNB-A and CNB-B domains. The vector was chosen from Arg230$^{RIIβ}$ in the PBC of CNB-A domain to Arg359$^{RIIβ}$ in the PBC of CNB-B domain (Fig 3D). The plots of φ, θ, and length of the vectors demonstrated clearly the different dynamics between the 2 CNB domains in the DnaJB1-PKAc and in WT PKAc RIIβ holoenzymes. In the RIIβ₂C₂ holoenzyme, the vector in one of the protomers moves and fluctuates significantly; the distance between domains varies from 50 to 70Å with a wide range of φ and θ angles movements (Fig 3E). The vector in the other protomer, however, is relatively less dynamic in both the φ and/or θ axis and length. The asymmetric dynamics of the CNB domains in each protomer of the RIIβ₂C₂ holoenzyme can also be observed here. In contrast, the vectors in both protomers of RIIβ₂J-C₂ holoenzyme remain relatively more stable. The plot of φ, θ, and length of the vectors in RIIβ₂J-C₂ holoenzyme showed less disperse angles

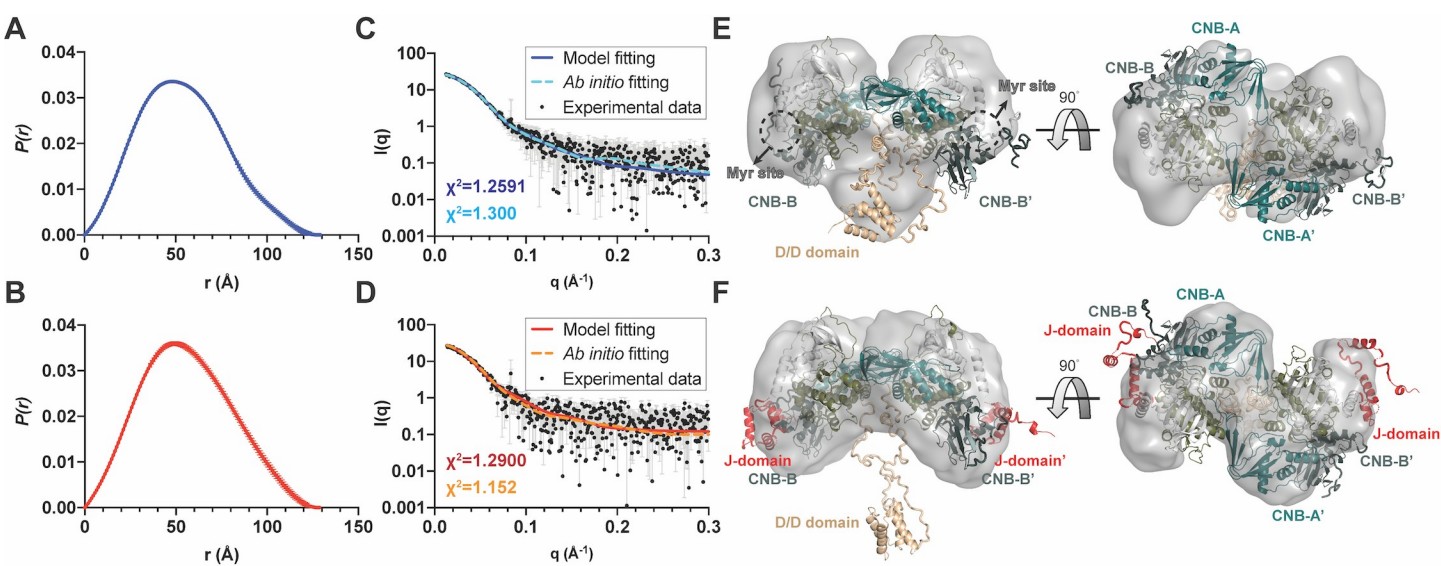

**Fig 4. SAXS analyses of RIIβ₂C₂ and RIIβ₂J-C₂ holoenzymes.** (**A, B**) $P(r)$ functions of RIIβ₂C₂ (A) and RIIβ₂J-C₂ (B) holoenzymes. (**C, D**) Scattering plots, atomic structure, and ab initio models fittings of RIIβ₂C₂ holoenzyme (C) with $\chi^2$ = 1.2591 (atomic structure) and $\chi^2$ = 1.300 (ab initio) and RIIβ₂J-C₂ holoenzyme (D) with $\chi^2$ = 1.2900 (atomic structure) and $\chi^2$ = 1.152 (ab initio). (**E, F**) The SAXS atomic structure models of RIIβ₂C₂ (E) and RIIβ₂J-C₂ holoenzymes (F) overlay with ab initio envelopes. The data used to make these figures can be found in S1 Data. CNB, cyclic nucleotide binding; SAXS, small-angle X-ray scattering.

and lengths movements (Fig 3F), and both vectors in the protomers populate a smaller φ, θ-space with less diverse length fluctuations. Asymmetric dynamics is also present in the RIIβ₂J-C₂ holoenzyme (Fig 3F). Together, this implies that the length of the N-terminus of the C subunit, whether longer as in the J-C subunit or shorter as in the C subunit used here, may have a differential impact on the observed asymmetric dynamics.

## SAXS analysis and localization of the D/D domain

The solution structures of both WT and DnaJB1-PKAc RIIβ holoenzymes were also determined by SAXS coupled with size-exclusion chromatography to support our cryo-EM structure. Based on pair distance distribution functions $P(r)$, both holoenzymes have similar dimension ($D_{max}$ = 129.09Å for WT holoenzyme and $D_{max}$ = 128.63Å for DnaJB1-PKAc holoenzyme). The $D_{max}$ values are consistent with our cryo-EM structure, where the 3D organization of RIIβ₂C₂ and RIIβ₂J-C₂ holoenzyme are similar (Fig 4A and 4B and Table 2). However, the DnaJB1-PKAc holoenzyme has a larger $R_g$ value than WT holoenzyme ($R_g$ = 41.69Å for

**Table 2. $R_g$ and $D_{max}$ values of RIIβ₂C₂ and RIIβ₂J-C₂ holoenzymes from SAXS.**

|  | RIIβ₂C₂ | RIIβ₂J-C₂ |
|---|---|---|
| $R_g$ (Å) from $P(r)$ | 41.69 ± 0.28 | 43.13 ± 0.41 |
| $D_{max}$ (Å) from $P(r)$ | 129.09 | 128.63 |
| Porod volume (Å³) from P(r) | 290,000 | 347,000 |
| $R_g$ (Å) from Guinier | 41.41 ± 0.49 | 43.29 ± 0.61 |
| MW estimation* (kDa) | 171 | 204 |
| Theoretical MW (kDa) | 173 | 187 |

*MW estimation was obtained by Porod volume/1.7.

MW, molecular weight; SAXS, small-angle X-ray scattering.

WT holoenzyme and $R_g$ = 43.13Å for DnaJB1-PKAc holoenzyme). A similar trend of $R_g$ values can also be obtained from Guinier analyses ($R_g$ = 41.41Å and $R_g$ = 43.29Å for WT and DnaJB1-PKAc holoenzyme, respectively) (S7A and S7B Fig and Table 2). The higher molecular weight of DnaJB1-PKAc can explain why RIIβ$_2$J-C$_2$ holoenzyme has a larger radius of gyration.

The higher molecular weight and the presence of the J-domain were also reflected in the larger Porod volumes of DnaJB1-PKAc holoenzyme (290,000Å$^3$ for RIIβ$_2$C$_2$ versus 347000Å$^3$ for RIIβ$_2$J-C$_2$) (Table 2). The molecular weight can be estimated by dividing the Porod volume by 1.7 according to the method of Petoukhov and colleagues [19]. The estimated molecular weight of RIIβ$_2$C$_2$ was obtained as 171 kDa in comparison to its theoretical molecular weight of 173 kDa (Table 2). The estimated molecular weight of the RIIβ$_2$J-C$_2$ holoenzyme was 204 kDa, while its theoretical molecular weight is 186 kDa (Table 2). Considering that the flexible J-domain enhances the domain dynamics of the holoenzyme, the result is that it delocalizes over a larger volume [20]. This can explain why RIIβ$_2$J-C$_2$ holoenzyme has a larger deviation between estimated and theoretical molecular weights than WT holoenzyme.

We further analyze Kratky plots of these 2 holoenzymes, which can provide a way to assess the degree of flexibility within the scattering macromolecules. Kratky plot analyses of these 2 holoenzymes showed bell-shape peaks at low q; however, neither of them converges to the q-axis at high q (S7C and S7D Fig), indicating that both of the complexes are multi-domains proteins with flexible regions [20]. These results are consistent with our structures and MD simulations where we identified several dynamic/flexible domains as well as a flexible linker.

To fit the SAXS data, we used the WT crystal structure as a starting model and further considered the dynamic properties of each domain. The missing linkers were built as flexible poly-Gly chains, while the missing D/D domain was generated using homology models from an online protein structure prediction software program, I-TASSER [21]. Both of our holoenzyme models fit the experimental SAXS data well ($\chi^2$ = 1.2591 and $\chi^2$ = 1.2900 for WT and J-C holoenzyme, respectively) (Fig 4C and 4D and S7E and S7F Fig). In both of the holoenzyme structural models, the D/D domain of RIIβ is localized on the same face as the CNB-B domains and the myristylation (Myr) sites of the C subunit (Fig 4E and 4F).

The ab initio models of WT and fusion holoenzymes were generated from the SAXS scattering data using the program DAMMIN [22], and these 2 models reveal good quality of fitting to the experimental data with $\chi^2$ score 1.300 and 1.152 for WT and J-C holoenzyme, respectively. Both atomic structure and ab initio approaches show complementary results that have similar 3D shapes (Fig 4C and 4D). Besides the 3D shapes, the dynamic features of the holoenzymes were also observed from the ab initio models. Consistent with our cryo-EM structure and MD simulation data, the CNB-B domains in the WT holoenzyme ab initio model are not fully visible suggesting that they are flexible, whereas the CNB-B domains remain clear in the J-C holoenzyme ab initio model as they are stabilized by the J-domains. Similar to the SAXS atomic model, the WT holoenzyme ab initio model also reveals the general position of the D/D domain which tethers at the same face as the CNB-B domains. The fact that the D/D domain is missing in the J-C holoenzyme SAXS envelope suggests that the J-domain may also influence D/D domain dynamics.

Although the position of the D/D domain cannot be identified unambiguously in our cryo-EM structure due to its flexibility, the extra density in our structure nevertheless provides further supporting evidence for the general position of the D/D domain (S8A and S8B Fig). The first visible N-terminal residue in our RIIβ structure is Ile104$^{RIIβ}$, which is located at the hole formed by the 2 RIIβ:J-C protomers (S8A Fig). The extra density locates near the center of the RIIβ holoenzyme, close to Ile104$^{RIIβ}$ and extends along the central hole to the same surface where the CNB-B domains and J-domains are located (S8B Fig).

## Altered biochemical function of the DnaJB1-PKAc RIIβ holoenzyme

We next asked whether the fusion protein affects not only RIIβ holoenzyme dynamics but also its biochemical properties. Both our MD simulations and our cryo-EM structure suggest that 1 J-domain communicates with its adjacent CNB-B domain and interferes with the overall dynamic properties of the holoenzyme (Fig 2B, 2F and 2G), In addition, several studies have shown that the dynamic features of the CNB domains play a significant role in the allosteric activation of the RIα holoenzyme by cAMP [23–27] and pathogenic mutations in RIα cluster in the 2 CNB domains [28]. We thus compared cAMP activation of the holoenzyme formed with J-C to holoenzyme formed with WT C subunit.

Holoenzyme formed with the fusion protein was easier to activate than the WT holoenzyme ($EC_{50}$ = 285nM for C subunit versus $EC_{50}$ = 170nM for J-C subunit) (Fig 5C and Table 3). To investigate the importance of the J-domain in the RIIβ holoenzyme activation by cAMP versus the absence of the first 14 residues (exon 1), we engineered and expressed several deletion constructs (Fig 5A and 5B and S9 Fig). Each of the 4 extra helices in the J-domain was deleted sequentially, generating J-C(Δ1–13), J-C(Δ1–38), J-C(Δ1–54), and C(Δ1–14) deletion mutants. C(Δ1–14) is equivalent to J-C(Δ1–69) where the whole J-domain is deleted. All of these mutants expressed as stable proteins and formed holoenzymes with RIIβ. The deletion mutants were all easier to activate with cAMP, and all showed lower Hill coefficients compared to the WT RIIβ holoenzyme (Fig 5C and Table 3). The mutant that simply lacked the first exon, C(Δ1–14), which is the same as J-C(Δ1–69), also showed a reduced $EC_{50}$, indicating that the presence of the first exon (residues 1 to 14) is crucial for proper activation and allosteric regulation of the RIIβ holoenzyme. Moreover, the Hill coefficient for cAMP activation actually showed negative cooperativity once exon 1 was removed (Table 3). Deletion of the first 14 residues in the PKA C subunit as well as the addition of the J-domain not only changed CNB-B domain dynamics but also affected the finely tuned allosteric activation of the RIIβ holoenzyme by cAMP.

To investigate how exon 1 of the C subunit influence the asymmetric dynamics and cAMP activation of the RIIβ holoenzyme, Gaussian accelerated MD (GaMD) simulations on $RIIβ_2C_2$, $RIIβ_2J-C_2$, and $RIIβ_2C(Δ1–14)_2$ were carried out. Specifically, we ask whether it is the J-domain or the absence of the residues 1 to 14 that account for the differences. Using the radius of cAMP as a probe, we calculated the solvent-accessible surface area of the PBC pocket of the CNB-B domains in each of the complexes. Both $RIIβ_2J-C_2$, and $RIIβ_2C(Δ1–14)_2$ complexes have higher accessible surface area than the WT complex (Fig 5D), suggesting that it is easier for cAMP to dock into the PBC pockets, which will facilitate activation. To further analyze the cAMP accessibility in the PBC pocket, we measured the distance between the Cβ atom in $Ala360^{RIIβ}$ and the Cγ atom in $Leu351^{RIIβ}$ (Fig 5E). This distance in the RIIβ holoenzyme is 6.9Å, while the PBC pocket is more closed (4.9Å) once it binds cAMP (Fig 5E). We then calculated the distance populations in our simulations. Consistent with our solvent-accessible surface analysis, both the $RIIβ_2J-C_2$ and $RIIβ_2C(Δ1–14)_2$ complexes reveal more open states than the WT holoenzyme (Fig 5F). In both analyses, there is a noticeable asymmetry between protomers in the cAMP accessibility of the PBC pockets for the $RIIβ_2C(Δ1–14)_2$ and $RIIβ_2J-C_2$ holoenzymes, supporting our earlier observations of asymmetry. The full-length WT holoenzyme exhibits milder asymmetry, consistent with our hypothesis that the protomers are tightly coupled (Fig 5D and 5F and S10 Fig). These results may help to explain why the J-C holoenzyme formed with RIIβ, as well as the 1–14 deletion mutant of WT C, is easier to activate with cAMP, while also having reduced cooperativity as measured by the Hill coefficient, compared to the WT RIIβ holoenzyme.

To further explore the effect of exon 1 deletion versus the J-domain fusion on the A-helix stability, we analyzed the helical propensity of the A-helix in 3 different constructs, WT C-,

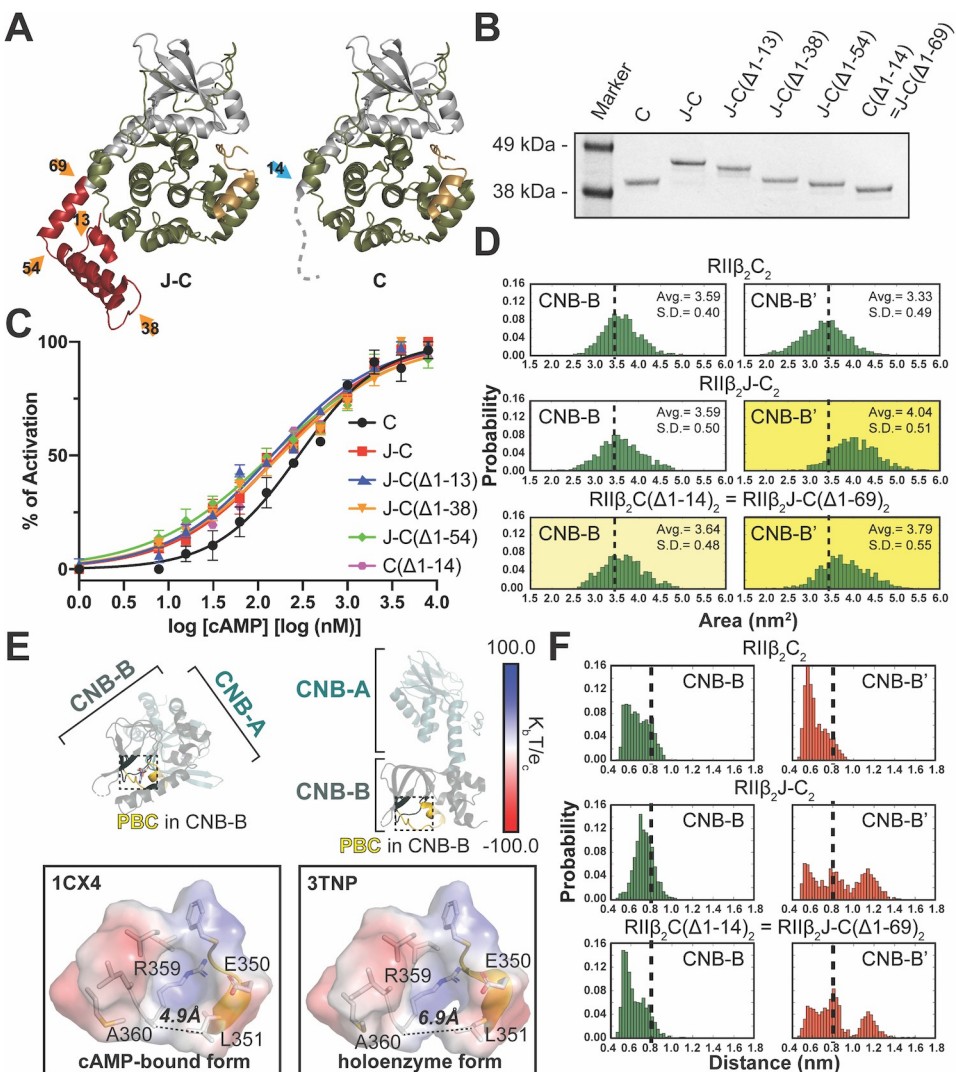

**Fig 5. The RIIβ$_2$J-C$_2$ holoenzyme is easier to activate with cAMP than the RIIβ$_2$C$_2$ holoenzyme.** (**A**) Structure of J-C subunit (left) and C subunit (right). The junctions of 4 helices in J-domain are labeled as orange arrows, where the first exon junction is labeled as blue arrow. (**B**) Coomassie blue staining SDS-PAGE of purified C-, J-C-, J-C(Δ1–13)-, J-C(Δ1–38)-, J-C(Δ1–54)-, and C(Δ1–14) subunits. C(Δ1–14) subunit is equivalent to J-C(Δ1–69) subunit, whole J-domain deletion. (**C**) The fusion protein RIIβ$_2$J-C$_2$ holoenzyme and its deletion mutants were easier to activate with cAMP than the WT RIIβ$_2$C$_2$ holoenzyme. $n = 2$ biological replicate. All error bars represent SEM. (**D**) The CNB-B domains in RIIβ$_2$J-C$_2$ and RIIβ$_2$C(Δ1–14)$_2$ holoenzymes have larger cAMP accessible surface area than RIIβ$_2$C$_2$ holoenzyme. (**E**) The distance between Cβ atom in Ala360$^{RIIβ}$ and Cγ atom in Leu351$^{RIIβ}$ in the CNB-B domain of cAMP-bound (left, PDB = 1CX4) and holoenzyme (right, PDB = 3TNP) forms. (**F**) The distance population probability of the PBC pocket in RIIβ$_2$C$_2$, RIIβ$_2$J-C$_2$, and RIIβ$_2$C(Δ1–14)$_2$ holoenzymes. The CNB-B domains in the RIIβ$_2$J-C$_2$ and RIIβ$_2$C(Δ1–14)$_2$ holoenzymes are more prone to open than the RIIβ$_2$C$_2$ holoenzyme. Both of the RIIβ$_2$J-C$_2$ and RIIβ$_2$C(Δ1–14)$_2$ holoenzymes reveal intrinsic asymmetry in the CNB-B domains. The data used to make these figures can be found in S1 Data. CNB, cyclic nucleotide binding; PDB, Protein Data Bank; SEM, standard error of the mean; WT, wild-type.

**Table 3. EC$_{50}$ and Hill coefficient of RIIβ$_2$C$_2$, RIIβ$_2$J-C$_2$ RIIβ$_2$J-C(Δ1–13)$_2$, RIIβ$_2$J-C(Δ1–38)$_2$, RIIβ$_2$J-C(Δ1–54)$_2$, and RIIβ$_2$C(Δ1–14)$_2$ holoenzymes from cAMP activation curves.**

|  | C | J-C | J-C(Δ1–13) | J-C(Δ1–39) | J-C(Δ1–55) | C(Δ1–14) |
|---|---|---|---|---|---|---|
| EC$_{50}$ (nM) | 285 ± 54 | 170 ± 32 | 144 ± 22 | 177 ± 31 | 141 ± 32 | 166 ± 38 |
| HillSlope | 0.949 ± 0.155 | 0.742 ± 0.091 | 0.749 ± 0.077 | 0.690 ± 0.074 | 0.644 ± 0.085 | 0.744 ± 0.119 |

J-C-, and C($\Delta$1–14) subunits (S11A Fig). The A-helix in the WT C subunit, which is part of the N-terminal linker that flanks the N- and C-lobes of the kinase core and is flanked by intrinsically disorder regions, has a surprisingly high helical propensity. Either deleting exon 1 or fusing with the J-domain decreases the predicted helical propensity (S11A Fig). Moreover, the last residue in the exon 1, Ser14$^C$, serves as a significant N-capping residue (S11B Fig). Decreasing the A-helix stability and missing the N-capping residue, as demonstrated in our computational and biochemical data, can influence the function and allosteric properties of the entire RII$\beta$ holoenzyme.

## Discussion

We describe here a structure of the RII$\beta_2$J-C$_2$ holoenzyme solved by cryo-EM single-particle 3D reconstruction. Although the overall compact 3D organization of the 2 RII$\beta$:J-C protomers in the holoenzyme structure is similar to the previous WT RII$\beta$ holoenzyme crystal structure, there are several differences in both the structure and dynamics in the DnaJB1-PKAc RII$\beta$ holoenzyme, which could result in the mis-regulation of PKA signaling. With cryo-EM, we are now able to observe more details about the dynamic features of the RII$\beta$ holoenzyme that could not be captured by conventional X-ray crystallography including the general localization of the D/D domain and the asymmetry of the CNB-B domains. Our structure together with MD simulations and biochemical studies not only allows us to better understand how DnaJB1-PKAc disrupts RII$\beta$ holoenzyme function and dynamics, but also makes us appreciate for the first time the intrinsic asymmetry of the RII$\beta$ holoenzyme that is likely an inherent feature of the activation mechanism that was masked in the earlier crystal structure. Our cryo-EM structure also confirms that the compact RII$\beta$ holoenzyme is distinct from the extended RII$\alpha$ holoenzyme.

### Structure and dynamics of WT and DnaJB1-PKAc RII$\beta$ holoenzymes

The cryo-EM structure and MD simulation data both reveal new and previously unappreciated features of the RII$\beta$ holoenzyme. First is the asymmetry of the RII$\beta$ holoenzyme, which is seen with MD simulations in the WT holoenzyme and captured in the cryo-EM structure but hidden in the previous crystal structure. With MD simulations, we observed an intrinsic asymmetry in the WT RII$\beta$ holoenzyme where 1 protomer has a more flexible CNB-B domain than the other. This same asymmetry is observed in the cryo-EM structure formed with the DnaJB1 fusion C subunit. The collective results suggest that the activation process most likely initiates from 1 protomer and then passes to the other. Does it then eventually spread to the entire holoenzyme, or does the intrinsic asymmetry simply involve toggling between the 2 protomers without resulting in full dissociation of the C subunits? Is the asymmetric feature an intrinsic part of the activation mechanism? How does the activation signal pass from 1 protomer to the other? These are the major future challenges. The cryo-environment and particle orientation problems of cryo-EM single-particle technique may limit us to visualize the full picture of protein dynamics; however, it still allowed us to capture an important conformational state of this holoenzyme. The beauty of single-particle cryo-EM, unlike crystallography, is that one can capture an ensemble of conformational states in a single experiment [29]; we anticipate that other variations on the compact globular conformation described here will be embedded in some of the other particles.

Second is the cross communication between the CNB-B domain with the J-Domain, which is revealed in the cryo-EM structure. As seen in previous structures, the J-domain is very flexible [11, 2]; however, in the RII$\beta$ holoenzyme, we see that the flexibility of the J-domain also appear to influence the dynamic features of its adjacent CNB-B domain as well as the A-helix

that it is fused to. The J-domain can also further influence the dynamic features of the CNB-B domains in the RIIβ holoenzyme, which we did not see in the RIα holoenzyme. Several studies have demonstrated that the dynamic features of the CNB domains in RIα are essential for PKA activation and allostery [25,26,30,31]. Here, for the first time, we are able to show that the RIIβ holoenzyme also has flexible CNB-B domains, although it is different from RIα in terms of its structure and allosteric regulation [32–34].

In summary, we show here that J-domain can influence the dynamic properties and the symmetry of the CNB-B domains. Our MD simulation data clearly indicated that the J-domain can stabilize the CNB-B domains, while the cryo-EM structure revealed how the J-domain and the CNB-B domain interact and together control the A-helix. We believe that this effect can be significant for stability, activation, cAMP binding, and/or interaction with other binding partners.

## D/D domain, myristylation site, and AKAP binding

Our cryo-EM model of the RIIβ holoenzyme, including the general position of the D/D domain, is consistent with our earlier crystal structure and SAXS data as well as with recent cross-linking data, and confirms that the compact RIIβ holoenzyme is quite different from the RIIα holoenzyme. The general positioning of the D/D domain, based on the low resolution cryo-EM density and SAXS, is consistent with the previous hypothesis that the unresolved N-terminal regions of RIIβ thread through the central hole of the RIIβ holoenzyme positioning the D/D domain close to the lower surface of RIIβ holoenzyme as seen in Fig 4 in close proximity to the CNB-B domains, the J-domain, and the bottom surface of the C-lobe in the kinase domain.

The D/D domain is essential for PKA localization. PKA holoenzymes, especially RII holoenzymes, typically bind with high affinity to an amphipathic helix that is embedded in scaffold proteins referred to as A-Kinase-Anchoring Proteins (AKAPs) through their D/D domains [35–37]. In this way, AKAPs bring the PKA holoenzyme to specific membrane locations in the cell where it is in close proximity to dedicated substrates such as receptors, transporters, and ion channels [5,38], and several lines of evidence have demonstrated that disrupting this interaction can affect not only PKA localization but also substrate specificity [38–43].

Our results suggest that the linker joining the D/D domain and the inhibitor site in RIIβ weaves through the hole that is created by the 2 RIIβ:J-C protomers. This placement of the D/D domain is consistent with cross-linking studies and different from the RIIα holoenzyme which was shown by negative staining EM to be extended [16,44]. It is also consistent with our earlier linker swap SAXS experiments showing that the motif that is responsible for the compact structure of the RIIβ holoenzyme is embedded in the linker [45]. This model places the D/D domain and the myristylation sites attached to the N-termini of the C subunits on the same general surface, which allows them to form multivalent interactions with membranes (Fig 6A). In solution, the myristyl groups in the RIIβ holoenzyme are solvent exposed and can anchor to nanodiscs even in the absence of AKAPs [42,46], whereas the myristyl groups in the RIα holoenzyme are embedded in the acyl pocket in the C-lobe of the C subunit and do not contribute to membrane anchoring in the absence of an AKAP [46]. In contrast to RIα holoenzymes, most of the RII subunits are always localized close to membranes [5]. Therefore, having the D/D domain in the RIIβ holoenzyme localized on the same face as the myristylation sites provides a dual mechanism for forming multivalent interactions with the membrane. It is also likely that anchoring the N-terminus to membranes could influence the conformation of the missing 14 residues. This segment with its basic residues followed by a phosphorylation site is in fact a classic myristylation motif, similar to the Src kinase, where basic residues also

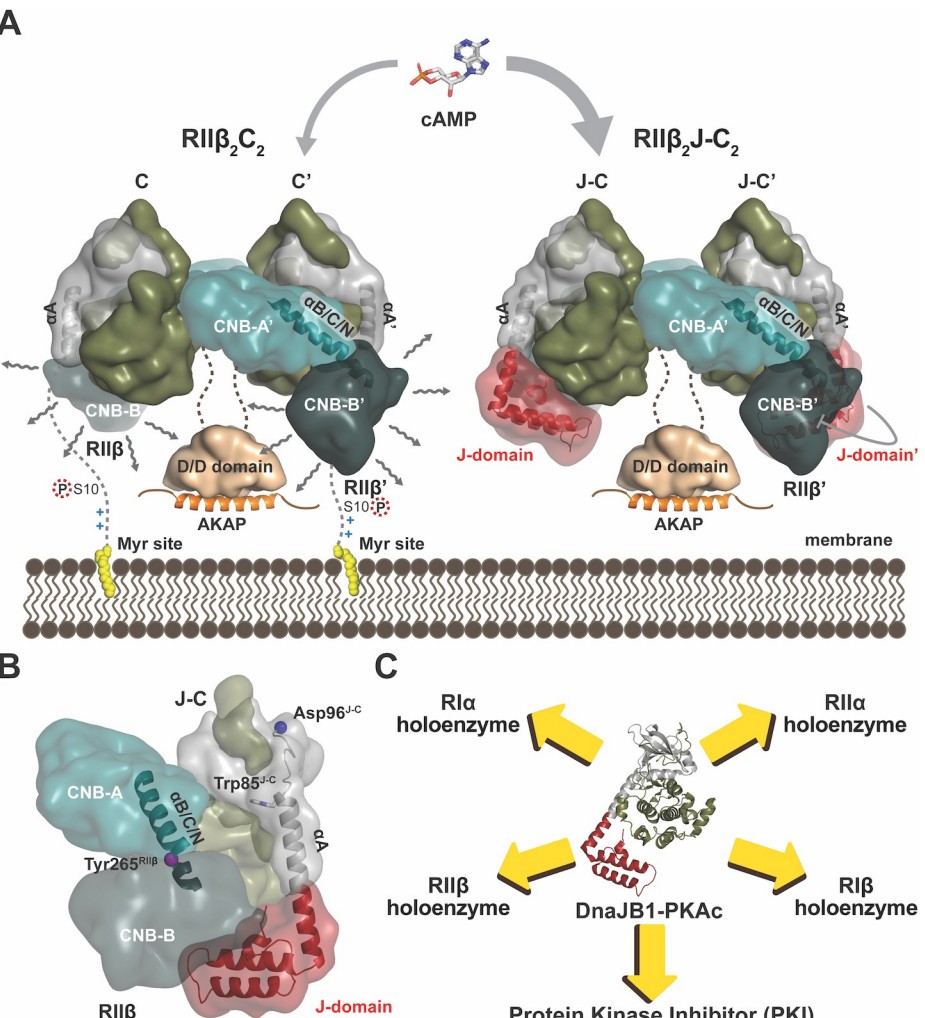

**Fig 6. The structural, dynamic, and allosteric features of RIIβ₂J-C₂ holoenzymes.** (**A**) The J-domain disrupts the multivalent interaction, and it can also stabilize the CNB-B domain dynamic in RIIβ holoenzyme. Meanwhile, RIIβ₂J-C₂ holoenzyme is easier to be activated by cAMP than RIIβ₂C₂ holoenzyme. (**B**) Two helices, B/C/N-helix and A-helix, regulate the RIIβ holoenzyme allostery. (**C**) DnaJB1-PKAc impairs PKA signaling network in an isoform-specific manner. CNB, cyclic nucleotide binding; PKA, cAMP-dependent protein kinase; PKI, protein kinase inhibitor.

contribute to membrane anchoring [47]. As we show here, ordering the first 14 residues in the membrane-bound holoenzyme could easily affect the dynamic behavior of the entire WT holoenzyme. The hydrophobic carboxyl terminus of the RIIβ subunit, as well as its dynamic β4-β5 loop in the CNB-B domain, could also become ordered when the holoenzyme is anchored to membranes. The highly basic β4-β5 loop in RIIβ is, in particular, in close proximity to the acidic surface of the membrane (Fig 6A). In addition, AKAPs themselves typically have a membrane-targeting motif (MTM), so many mechanisms could be used to stabilize interactions with membranes (Fig 6A). In the J-C fusion protein, the myristylation site is lost which could also affect, although not necessarily abolish, RIIβ holoenzyme binding to membranes in cells (Fig 6A).

## Two helices drive the allosteric regulation of the PKA holoenzymes

There are 2 helices that drive the allosteric regulation of PKA, the B/C/N-helix in the R subunits and the A-helix in the C subunit (Fig 6B). Sequences difference in the B/C/N-helix, as

well as differences in helical propensity and dynamics, between RI and RII reveal distinct allosteric networks between the 2 PKA isoforms (S12 Fig), and the hinge point position in the B/C/N-helix can also clearly contribute in unique ways to this allosteric communication. Our results indicate not only that the location of the major hinge point in the cAMP-bound conformation is significant, but also suggests that the B/C/N-helix dynamics and the activation processes are probably linked (S12 Fig).

While the dynamic B/C/N-helix of each R subunit is a dominant feature for cAMP-mediated activation of each holoenzyme, the stable A-helix is a dominant allosteric feature of the C subunit. The A-helix is embedded in the N-terminal tail (N-tail) (residues 1 to 39) that wraps around both lobes of the kinase core (Fig 6B). This N-tail that precedes the kinase core and is missing entirely in our cryo-EM structure is a key allosteric regulatory element [48]. Of particular importance is the hydrophobic motif (Trp30$^C$ and Phe26$^C$ or Trp85$^{J-C}$ and Phe81$^{J-C}$) at the end of A-helix, which in the WT C subunit, is wedged between the C-helix and the activation loop of the kinase core (Fig 6B)[48].

The importance of exon 1 and the A-helix in the PKA C subunit has been demonstrated both biochemically and computationally [49,50]. Biochemically, we show that the deletion of residues 1 to 14 (exon 1) in WT C subunit can introduce instability and can also affect holoenzyme function [49]. Our previous study also indicated that the J-C subunit is less thermostable than WT C subunit [11,12]. Several posttranslational modification sites in the WT C subunit are localized at the other end of the A-helix in exon 1, and these are missing in the fusion protein. These include the myristylation site at Gly1$^C$, the phosphorylation at Ser10$^C$, and deamidation at Asn2$^C$, and all are thought to contribute to function [51–53]. How these residues are ordered when the acyl group is anchored to a membrane and how this influences the structure and function of the holoenzyme remains to be elucidated. Computationally, the community map analysis based on the MD simulations shows that the DFG motif, which is crucial for kinase activity, is in the same community with the A-helix [50]. This community also connects to other substrate binding motifs, indicating these motifs are allosterically coupled [50]. It is clear that the A-helix can regulate many other motifs in both the C and R subunits, so that destabilizing the A-helix could have a major effect on holoenzyme function.

## Visualizing allostery

By capturing full-length PKA holoenzymes in a crystal lattice, we were able to visualize the striking symmetry of PKA as well as the allosteric cross talk between the 2 protomers, which is an essential feature of PKA activation [7,8,10,11]. However, any differences in dynamics, in particular of the CNB-B domains, are masked in these holoenzyme crystal structures most likely by crystal packing. We show here that the asymmetry, although hidden in the RIIβ crystal structure, can be seen with MD simulations and in the cryo-EM structure. In the crystal structure, the enhanced dynamics and/or flexibility is reflected in high temperature factors for the CNB-B domains and the A-helix relative to the CNB-A domain ant the rest of the C subunit, but the asymmetry, which most likely is an integral part of the allosteric mechanism for activation, is hidden. MD simulations allowed us to delve more deeply into the intrinsic asymmetry that is embedded in the dynamic properties of each CNB domain. This loss of allosteric communication between the 2 protomers is also reflected in the reduced Hill coefficient for cAMP activation. With single-particle cryo-EM reconstruction, however, we can directly observe the asymmetry of the 2 protomers. The combination of crystallography, MD simulations, and cryo-EM is thus a powerful way to explore the different states; 1 approach alone is not sufficient.

## DnaJB1-PKAc disrupts PKA function in an isoform-specific manner

PKA-dependent signaling is finely regulated, which is demonstrated best in RIα where there are many mutations that are mostly clustered in the 2 CNB domains [28]. Those in the CNB-A domain, for example, lead to Carney complex disease and create holoenzymes that are easier to activate with cAMP, while those in the CNB-B domain lead to acrodysostosis and are more difficult to activate [28]. The differences are not off/on, but simply reflect a slightly different balance which is sufficient to cause distinct disease phenotypes.

Although the DnaJB1-PKAc fusion protein is the driver of FL-HCC, our structures of the RIα holoenzyme and the RIIβ holoenzyme with J-C subunit do not reveal a clear mechanism for the pathogenesis of the mutation; instead, we showed that the regulation of each holoenzyme is disrupted. The RIIβ holoenzyme formed with the fusion protein is easier to activate with cAMP than WT holoenzyme. This is in contrast to RIα where the PKAc and DnaJB1-PKAc RIα holoenzymes show similar cAMP activation [11]; however, DnaJB1-PKAc can disrupt RIα localization and cAMP signaling compartmentation [54]. Each of these PKA signaling networks is highly regulated, and perturbation of that fine tuning can have profound consequences. In contrast to mutations in the R subunits, the results of a mutation or fusion in the C subunit will be complex and multivalent, and every holoenzyme will be affected as well as interactions with PKI (Fig 6C). In the liver where regulation of metabolism is linked so closely to PKA-mediated gene transcription, the effects of this mutation will be complex. Based on our work here, coupled with previous studies of the PKI complex and the RIα holoenzyme, it is clear that the fine tuning and allosteric regulation of each complex will be altered as well as the expression levels of each PKA regulatory subunit; the entire PKA signaling network will be disrupted.

## Supporting information

**S1 Fig. Negative stain EM confirms that RIIβ₂J-C₂ and WT RIIβ holoenzymes have similar architectures.** (**A**) Representative micrograph for negatively stained RIIβ₂J-C₂ holoenzyme. Example particles are shown in red boxes. (**B**) 2D class averages of RIIβ₂J-C₂ shown alongside projections of WT RIIβ holoenzyme crystal structure. The WT RIIβ₂C₂ crystal structure was filtered to 20Å. (**C**) Model and negatively stained EM density of RIIβ₂J-C₂ holoenzyme. EM, electron microscopy; WT, wild-type.
(TIF)

**S2 Fig. Cryo-EM structure of C2 symmetric RIIβ₂J-C₂ holoenzyme.** (**A**) Representative tilted micrograph with the right half of the image showing picked particles. (**B**) Local CTF plot for micrograph in (A). Image generated using Appion and Gctf. (**C**) Representative 2D class averages. (**D**) 3D FSC curve. The data used to make this figure can be found in S1 Data. cryo-EM, cryo-electron microscopy; CTF, contrast transfer function; FSC, fourier shell correlation.
(TIF)

**S3 Fig. 3D classification of RIIβ₂J-C₂ holoenzyme.** (**A**) 3D classification. (**B**) Local density map of RIIβ₂J-C₂ holoenzyme.
(TIF)

**S4 Fig.** Temperature factor of RIIβ2C2 holoenzyme (PDB = 3TNP). Both the A-helix and the CNB-B domain reveal high temperature factor. PDB, Protein Data Bank.
(TIF)

**S5 Fig. The comparison of RIα and RIIβ B/C/N-helices.** (**A**) The B/C/N-helices of RIα and RIIβ have different hinge angles. (**B**) The B/C/N-helix of RIα have high helical propensity with

a local minimum at Gly235$^{RI\alpha}$. In the RIIβ, helical propensity of B/C/N-helix is much lower with a break point at Tyr265$^{RIIβ}$. (**C**) N-capping analysis of B/C/N-helix in the RIα and RIIβ. (**D**) C-capping analysis of B/C/N-helix in the RIα and RIIβ. The data used to make these figures can be found in S1 Data.
(TIF)

**S6 Fig. The overlaid of all states of each RIIβ subunit protomer in RIIβ$_2$J-C$_2$ holoenzyme from MD simulations.** Both CNB domains show similar dynamics and have breakages at Tyr265. Residue Tyr265$^{RIIβ}$ was shown as pink ball. CNB, cyclic nucleotide binding; MD, molecular dynamics.
(TIF)

**S7 Fig. SAXS analyses of RIIβ$_2$C$_2$ and RIIβ$_2$J-C$_2$ holoenzymes.** (**A**, **B**) Guiner plots of RIIβ$_2$C$_2$ (A) and RIIβ$_2$J-C$_2$ holoenzymes (B). (**C**, **D**) Kratky plots of RIIβ$_2$C$_2$ (C) and RIIβ$_2$J-C$_2$ (D) holoenzymes both show bell-shape peaks at low q and not converging to the q-axis at high q. (**E**, **F**) Scattering plots at low q and the model fittings of RIIβ$_2$C$_2$ (E) and RIIβ$_2$J-C$_2$ (F) holoenzymes. The data used to make these figures can be found in S1 Data. SAXS, small-angle X-ray scattering.
(TIF)

**S8 Fig. Cryo-EM structure of RIIβ$_2$J-C$_2$ holoenzyme reveals the general position of D/D domain.** (**A**) The extra density near the residue Ile104$^{RIIβ}$ locates at the central hole of RIIβ$_2$J-C$_2$ holoenzyme. Residues Ile104$^{RIIβ}$ were labeled as yellow balls. (**B**) The extra density extends along the central hole to the same face as the CNB-B domains and J-domains. Residues Ile104$^{RIIβ}$ were labeled as yellow balls. CNB, cyclic nucleotide binding; cryo-EM, cryo-electron microscopy; D/D, Dimerization/Docking.
(TIF)

**S9 Fig. Full coomassie blue staining SDS-PAGE of purified C-, J-C-, J-C(Δ1–13)-, J-C(Δ1–38)-, J-C(Δ1–54)-, and J-C(Δ1–69) subunits.** J-C(Δ1–69) subunit is equivalent to C(Δ1–14) subunit.
(TIF)

**S10 Fig. Displacement vector between CNB-A (Arg230$^{RIIβ}$) and CNB-B (Arg359$^{RIIβ}$) domains.** Each pair of panels shows the 2 protomers (left and right) in the holoenzyme. Plots show θ(x), φ(y), and (d) distance in color. Each row is an independent simulation. Upper left panels are conventional MD, as shown in Fig 3 of the main text, and other panels are from GaMD. Full-length C (C-gamd) shows on average a muted asymmetry between protomer displacement vectors. Asymmetry (Δθ,Δφ,Δd) is measured as the average per frame difference of θ, φ, and d between protomers, calculated as $\Delta X = 100 * \frac{\sum_{i,j}^{N}(Xi/\overline{Xi} - Xj/\overline{Xj})}{N}$, where $i$ and $j$ are protomers at matching frames. The data used to make these figures can be found in S1 Data. CNB, cyclic nucleotide binding; GaMD, Gaussian accelerated MD; MD, molecular dynamics.
(TIF)

**S11 Fig. Helical propensity analyses of C subunit, J-C subunit, and C(1–14Δ).** (**A**) Either fusing with J-domain or just simply deletion of first exon reduces the helical propensity of A-helix. (**B**) Capping propensity analysis of the A-helix. Ser14$^C$ reveals a strong N-capping propensity. The data used to make these figures can be found in S1 Data.
(TIF)

**S12 Fig. Isoform-specific PKA structures and B/C/N-helix dynamics.** PKA RIα and RIIβ undergo different conformational changes between holoenzyme form and cAMP-bound form.

PKA, cAMP-dependent protein kinase.
(TIF)

**S1 Methods. Materials and methods.**
(DOCX)

**S1 Raw Images. Full coomassie blue staining SDS-PAGE of purified C-, J-C-, J-C(Δ1–13)-, J-C(Δ1–38)-, J-C(Δ1–54)-, and J-C(Δ1–69) subunits.** J-C(Δ1–69) subunit is equivalent to C(Δ1–14) subunit.
(PDF)

**S1 Data.**
(XLSX)

## Acknowledgments

We thank all the members in Taylor and Cianfrocco laboratories. We also thank all the staffs/scientists at ALS for the help with beam access and data collection/analysis.

## Author Contributions

**Conceptualization:** Tsan-Wen Lu, Michael A. Cianfrocco, Susan S. Taylor.

**Data curation:** Tsan-Wen Lu, Phillip C. Aoto, Jui-Hung Weng, Cole Nielsen, James Hall, Ping Zhang, Michael A. Cianfrocco.

**Formal analysis:** Tsan-Wen Lu, Phillip C. Aoto, Jui-Hung Weng, Cole Nielsen, Michael A. Cianfrocco, Susan S. Taylor.

**Funding acquisition:** Michael A. Cianfrocco, Susan S. Taylor.

**Investigation:** Tsan-Wen Lu, Michael A. Cianfrocco, Susan S. Taylor.

**Methodology:** Tsan-Wen Lu, Phillip C. Aoto.

**Project administration:** Tsan-Wen Lu.

**Resources:** Sanford M. Simon.

**Supervision:** Michael A. Cianfrocco, Susan S. Taylor.

**Validation:** Tsan-Wen Lu, Phillip C. Aoto, Jui-Hung Weng, Jennifer N. Cash, Michael A. Cianfrocco.

**Visualization:** Tsan-Wen Lu, Phillip C. Aoto, Jui-Hung Weng, Ping Zhang, Michael A. Cianfrocco, Susan S. Taylor.

**Writing – original draft:** Tsan-Wen Lu, Michael A. Cianfrocco, Susan S. Taylor.

**Writing – review & editing:** Tsan-Wen Lu, Phillip C. Aoto, Jennifer N. Cash, Sanford M. Simon, Michael A. Cianfrocco, Susan S. Taylor.

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
