## [Editor Report · Decision Letter 0]

26 May 2020

Dear Susan, 

Thank you for submitting your manuscript entitled "Capturing allostery: asymmetry and dynamics of PKA RIIβ holoenzyme with DnaJB1-PKAc fusion in fibrolamellar hepatoceullar carcinoma" for consideration as a Research Article by PLOS Biology. I sincerely apologize again for the delay due to COVID-19-related disruptions.

Your manuscript has now been evaluated by the PLOS Biology editorial staff as well as by an Academic Editor with relevant expertise and I am writing to let you know that we would like to send your submission out for external peer review.

Please re-submit your manuscript within two working days, i.e. by May 28 2020 11:59PM.

Kind regards,

Hashi Wijayatilake, PhD,

Managing Editor

PLOS Biology

---

## [Decision Letter · Decision Letter 1]

15 Jul 2020

Dear Susan,

Thank you very much for submitting your manuscript "Capturing allostery: asymmetry and dynamics of PKA RIIβ holoenzyme with DnaJB1-PKAc fusion in fibrolamellar hepatoceullar carcinoma" for consideration as a Research Article at PLOS Biology. Thank you also for your patience as we completed our editorial process, and please accept my apologies for the delay in providing you with our decision. Your manuscript has been evaluated by the PLOS Biology editors, an Academic Editor with relevant expertise, and by four independent reviewers.

As you will see, the reviewers are generally enthusiastic about your study, however they also have some concerns that need to be addressed. Reviewer 2 raises some questions about the very large numbers of particles that have been discarded during the processing. Reviewer 3 asks for an additional MD simulation of the C-subunit with its N-terminal 14 residues, whereas Reviewer 4 requests that you tone down your conclusions about the D/D location, which s/he finds are not supported by the data. In addition, the presentation of Figure 1 should be improved.

In light of the reviews (attached below), we will not be able to accept the current version of the manuscript, but we would welcome re-submission of a much-revised version that takes into account the reviewers' comments. We cannot make any decision about publication until we have seen the revised manuscript and your response to the reviewers' comments. Your revised manuscript is also likely to be sent for further evaluation by the reviewers.

We expect to receive your revised manuscript within 2 months. 

**IMPORTANT - SUBMITTING YOUR REVISION**

*Re-submission Checklist*

*Published Peer Review*

*PLOS Data Policy*

*Blot and Gel Data Policy*

Sincerely,

Ines

--

Ines Alvarez-Garcia, PhD

Senior Editor

PLOS Biology

Carlyle House, Carlyle Road

Cambridge, CB4 3DN

+44 1223–442810

Reviewers’ comments

Rev. 1:

This is a nice study that uses a range of biophysical and structural techniques to probe the regulation mechanism of PKA. The study picks up from the group's previous crystallographic work which found a symmetric dimer of the RIIb2C2 holoenzyme, and using cryoEM for a fusion associated with fibrolamellar hepatocellular carcinoma (DnaJB1-PKAc) found that the holoenzyme might actually be asymmetric. Molecular dynamics shows that the fusion holoenzyme is relatively more stable than the wild-type, which displays increased asymmetry. A SAXS study is conducted which can be improved. Their enzymatic studies imply that asymmetry plays a role in regulation of activity, but as they rightly highlight in the discussion, the current study represents a first step towards understanding exactly what is occurring here. I have some specific comments that they might wish to address.

Comments

Figure 1A is not the clearest. They should illustrate which regions of PKA and DnaJB1 are fused together, and they might also wish to provide a schematic that describes the structures in the figure. Similarly, the asymmetry that they observe in their cryo map is perhaps not as easy to visualize in Figure 1D as it is in Fig 3B, perhaps they might consider promoting that panel to the main body.

The hinge angles in Figure 2F and 2G are curiously accurate to 2 decimal places. I suspect the crystal structures provide an accuracy approximating to a few degrees, and would encourage the authors to take another look at these.

The SAXS studies can be significantly improved. These should be conducted over a range of concentrations to validate the differences observed. They should be presented in a manner described by Trewhella PMID: 28876235, and a full SAXS 'Table 1' presented. Figures 4E and F are uninformative without the envelopes. The spatial localization of the D/D domain from the SAXS analysis is unconvincing. Does the MW difference hold up in I-zero analysis?

It is unclear how many repeats have been conducted for the enzymatic assays in Fig 5, and what the error bars indicate.

Rev. 2:

This study focuses on the structure and dynamics of PKA holoenzymes containing RIIbeta and DNAJB1-PKAc subunits, and includes a number of interesting observations. The authors utilise cryo-EM in one of the first (the first?) attempts at a high-resolution 3D reconstruction of a PKA holoenzyme using cryo-EM. Clearly, PKA is clearly not a straightforward sample for analysis by cryo-EM, and the masking approach employed to resolve the CNB-D and J domains is smart and creative. The structure reveals unexpected potentially asymmetric dynamics in the CNB-B/J domain regions of the holoenzyme. The study includes a very extensive investigation of IIbeta holoenzyme dynamics using MD simulations that supports the notion that dynamics are restricted to one RII protomer at a time. Another notable observation is electron density in the cryo-EM structure that helps to assign the position of the D/D domain relative to the rest of the holoenzyme. The study pulls together several complementary techniques (cryo-EM, SAXS, MD simulations, PKA activation assays), is well written, and the methods are described in detail.

I have a number of minor comments/questions:

1. In the abstract on lines 39-41, I was confused by the following statement: "properties of the RIIβ holoenzyme are also impaired by the J-domain; it is harder to activate with cAMP and the cooperativity is reduced". The data in Fig 5C show that the mutant holoenzyme is easier to activate with cAMP?

2. Regarding the legend to figure 1. Can the authors clarify which PDB coordinate file is being shown in each panel? It isn't immediately apparent.

3. Regarding the rotations between panels in figure 1. The rotation operations between panels are unnecessarily complicated - if the left and middle views are switched then a simple 90 degree rotation between each panel will be sufficient.

4. Regarding the cryo-EM processing (table 1 and methods). I notice that very large numbers of particles have been discarded during the processing (down from 642,843 to 69,605 for the C2 structure and 11,182 for the C1 structure), more so than in a typical cryo-EM reconstruction. Please can the authors comment on why this was necessary?

5. Regarding how this study relates to other structural studies of PKA that used EM. The intro does not refer to other EM studies. I am aware of one paper from the Scott/Gonen labs that used negative stain EM (E-life, 2013, PMID 24192038). This study focused on type II PKA-AKAP18gamma complexes, and the authors reported several extended conformations in which the two RII-C protomers were separated by some distance. This study does not include any cryo-EM experiments, however, and does not attempt to quantify the abundance of different conformational states. Did the authors of this study observe any extended conformations of PKA?

6. Regarding the novelty of the cryo-EM reconstruction. There do not appear to be any other published 3D reconstructions of PKA using cryo-EM. Is this correct? It would be helpful to include an additional statement in the intro or discussion explaning how this work relates to other EM investigations of PKA.

7. Regarding the issue of asymmetry as detected in the cryo-EM reconstruction. In the initial C2 averaged structure (using ~ 69K particles), the CNB-B and J domains are not visible. By applying a mask and utilizing only a subset of the particles (~11K in the final refined structure), the authors are able to resolve the CNB-B and J domain in one half of the holoenzyme. They conclude that this shows asymmetry in the holoenzyme with one RII-JC promoter more ordered than the other. Is it possible that this apparent asymmetry has arisen due to selective sampling of the particle images? Ie the same conformation is in fact present at the same frequency in both protomers (ie at a frequency of about 1 in 7) such that this conformation can only be resolved in the protomer on which the averaging is focused? Please can the authors comment on this possibility? A statement should be included in the main text at least considering this possibility. Perhaps it is also possible to try to further classify the 11,182 particles to look for a subset that show the same ordered conformation in both protomers?

8. On lines 444-445. "However, any differences in dynamics are masked in these holoenzyme crystal structures; they are simply averaged". Is it not more likely that crystal packing prevents these movements vs averaged out during data processing?

9. In supplementary figure 12, the legend refers to coomassie-stained gels that are absent in the figure (they appear to have been moved to figure 5A). Please can the authors correct this.

Rev. 3:

This interesting paper uses a combination of CryoEM and MD models to examine the structure of DNAJB1-PKACA (J-C) fusion protein: RII� tetrameric holoenzyme. The J-C fusion is caused by a genomic translocation which drives a rare form of liver cancer and is worthy of study. The data presented are interesting and shed light on both the impact of fusion (biochemical and structural) and also potentially on the behaviour of the endogenous PKA RII�� holoenzyme. Unlike their previous PKA RII��holoenzyme crystal structure the D/D domain is visible in the CryoEM, which adds value to the new structure. Also they report intrinsic asymmetry with two J-C protomers in the tetramer and suggest this may be important for regulation of the holoenzyme. There are however a couple of issues which should be addressed to clarify interpretation, particularly with regard to extrapolating findings to the endogenous PKA complex.

Major Issue:

1. Fusion vs truncation. The paper focuses on the structure of the J-C fusion which is valuable. Comparisons are made with the C subunit alone, but the studies often compare the J-C fusion with the C subunit without the N-terminal 14aas (as they are deleted in the fusion). In particular, the MD dynamics do not include this 14aa section of the C subunit. This is important as it is not clear whether many of the observations made are because of deletion of the 14aa segment OR because of fusion with the J-subunit. Linked to this, the authors often discuss how this holoenzyme structure - and in particularly asymmetry - could be crucial to holoenzyme function. It should perhaps be made clearer that the current structure has a mutant catalytic subunit, which may of course behave differently to the native form with no fusion and with no truncation. It is understood that generating a Cryo-EM structure of the native holoenzyme for comparison may be a huge undertaking. However, as a minimum, the MD analysis could be conducted on the C-subunit with the inclusion of the N-terminal 14 aas. Perhaps with the FL-C subunit, the complex will behave differently.

2. This main point is particularly important given the biochemistry in Figure 5 , showing that deletion of the 14aa N-terminus has a similar impact on cAMP binding to addition of the J-domain. This implies that the J-domain may be dispensable for the alteration to biochemical function. Related to this, referring to the C�1-14 as RIIβ2J-C(�1-69)2 is a bit misleading (it contains none of the J domain). Although the fact that they are the same thing is made clear, through the text it should either be referred to as C�1-14 or given both descriptions.

3. In the text, differences in the B/C/N helices between protomers in the Cryo structure are discussed - could these differences be made more clear in a figure as they are somewhat lost in the images of the whole structure? (Lines 133 and 134)

4. The authors state the holoenzyme orients in a particular way on the cryo EM grid. Might attachment to the grid affect the dynamics of the two distinct protomers, direct the asymmetry or impact on the described loss of density in some regions? I acknowledge that the MD analysis provides an orthogonal approach.

5. Line 83: "To understand the pathological consequences of FL-HCC, we used single particle cryo-electron microscopy (cryo-EM) to determine a structure of the RIIβ holoenzyme formed with J-C." This sentence is a bit misleading. Do the authors mean to understand the role of the J-C fusion in FL-HCC pathology?

6. Minor: In the introduction, the authors note that in FL-HCC, RIa is upregulated and RIIb is downregulated. This could imply that the stability or biochemistry of the complex may result in compensatory changes in subunit expression. This may also suggest that the J-C is preferentially complexed with the RI subunit in tumours (rather than as part of the RIIb structure presented here). Could this be commented on? Are there data on the stability or half life of distinct J-C holoenzymes?

Linked to this, above, interaction between J and CNB-B, results in apparent CNB-B stabilisation. Given the origin of this chimeric protein, such interactions are clearly not evolved but rather are interactions driven by their chance proximity (resulting from the in frame translocation). This is perhaps not given clear consideration in the text although the differential with RIa is covered.

Rev. 4: William Rice – note that this reviewer has waived anonymity

Introduction

This paper examines the structure of the fusion protein J-C subunit bound to RIIβ. The JC subunit is created when the J domain of Dna JB1 replaces exon 1 of cAMP dependent protein kinase (PKA), and it is a potent driver of Fibrolamellar heptatocellular carcinoma. The authors used a combination cryo-EM, SAXS, and molecular dynamics to produce moderate resolution (6-8 Å) structures of the holoenzyme and examine its structural dynamics. Although they do not reach high resolution, they were able to identify various domains of the protein and compare it with a prior crystal structure of the wild-type construct. This will be of interest to the many groups on the field studying PKA activation and cancers related to the overactivation of this protein. The dynamical results will inspire experiments to more fully understand the activation and mechanism of this complex. I believe it is of enough general interest to publish in PLOS Biology.

Summary of Results

The holoenzyme has stoichiometry RIIβ2J-C2. The cryo-EM structure of the holenzyme itself was only of moderate resolution, 6.5 Å, from 69,605 particles with C2 symmetry applied. Much of the mass was missing, including the J domains, cyclic nucleotide B (CNB-B) domains, dimerization and docking (D/D) domains were missing. Further analysis using 3D classification and removing C2 symmetry followed by focused classification revealed a subset of 11,182 particles which did show the J and CNB-B domains in one of the subunits of the holoenzyme, at 7.5 Å resolution. The authors conclude that these regions must be flexible, there must be asymmetry between the protomers in these domains, and presumably both are important for the function of this protein. Molecular models were made from the maps using previously published PDB coordinates and homologous model structure prediction, followed by real space refinement.

Molecular dynamic simulations of RIIβ with wt C subunits indicated that the C subunit was stable, as was the CNB-A domain of the RIIβ in one of the protomers. In contrast, the CNB-B in the other protomer appeared to have a large amount of flexibility, in agreement with the asymmetry seen in the cryo-EM map of RIIβ2J-C2. The pivot point appears to be at Tyr265, located in the BCN helix of RIIβ. The same simulations performed on the RIIβ2J-C2 holoenzyme showed that the CNB-B subunits of both protomers had a similar amount of flexibility, and both were more stable than the flexible CNB-B subunit of the wt form. Both J domains were overall more flexible than the remainder of the C domain. They hypothesize that the J domains are coupled to the CNB-B domains, due to their apparent close proximity in the cryo-EM map. Analysis of the flexibility between the CNB-A and CNB-B domains also showed greater flexibility in one protomer than the other for both wt and RIIβ2J-C2 isoforms, though flexible wt protomer moved more than the flexible RIIβ2J-C2 protomer. Further MD simulations using GaMD measured the solvent accessible surface of RIIβ2C2, RIIβ2J-C2, and RIIβ2J-C(Δ1-69)2. Both constructs had more solvent accessible area than the wild type, and distance measurements between two atoms in the PBC pocket also showed more open states than seen in the wt complex. There was more asymmetry observed between the protomers than seen in the wt complex.

SAXS analysis of wt and RIIβ2J-C2 holoenzymes showed similar results for both, with indications of stably folded domains and a high degree of flexibility. The RIIβ2J-C2 isoform showed a slightly higher radius of gyration and a higher deviation of predicted versus calculated molecular weight. Finally, they measured activation of wt and fusion holoenzymes by cAMP, as well as successive deletion mutants of the RIIβ2J-C2 where successive N-terminal helices of the J domain were removed. Apart from the wt, all fusion holoenzymes showed a higher affinity for cAMP and lower cooperativity between subunits.

Based on these results, and prior data in the literature, a model for J-domain activation is presented.

Review

This is a significant paper in that the moderate resolution structure of an ocogenic fusion protein was solved. Likely due to its inherent flexibility, it only went to 6.5 Å but further processing revealed a previously unseen asymmetry in the protomer. This led to molecular dynamics simulations, which also predict an asymmetry on the protomer. The structure came from 1,129 micrographs, which is a moderate dataset nowadays. Adding more images would likely improve the resolution, but given the flexibilities involved, likely not enough to significantly alter the results. The most significant part of the structural interpretation which seems to not be on solid ground is the D/D location, discussed below:

D/D Domain

Results Lines 262-273: The positioning of the D/D domain due to the “extra density” in the map seems speculative and should be in the Discussion part of the paper. Although there is other experimental evidence such as crosslinking which points to its localization, the extra density seen in the map is at best very low resolution and it is hard to justify using it to place atoms. Considering the moderate resolution of the rest of the map, fitting anything in this poorly resolved density seems unjustified. There is no evidence in the cryo-EM map to place the D/D domain in close proximity to the CNB-B and J-domains. This could be moved to the discussion section in interpreting the map. The extra density could be an artifact due to low resolution, though it could be the linking region. As shown in Supplemental Figure 9, it does not seem to match two unstructured loops. The cutoff value for the density used in Figure 9 also does not match the J/A helix well (left side of Figure B) but there is no extra protein proposed to fill this region.

Discussion line 337: The cryo-EM map does not give the general localization of the D/D domain. It is not seen, there is only some undefined density above the proposed location which the authors propose to be a linker to this domain. This should not be a primary conclusion of this paper.

Discussion line 381: The results do not show that the linker joining the D/D domain weaves through the hole in the dimer. As mentioned above, this is conjectured but not shown by the observed extra density. The conclusion is made too strongly given the data at hand. It is OK to hypothesize the location but the maps do not definitively find it.

In short, the conclusions about this domain should be made much less strong. Stronger evidence would be required to justify these conclusions. This could possibly come from further analysis/classification of the existing data, but would likely need more data to be added, if it could be resolved at all. I do not recommend more data, unless it is already available or the authors are willing, but in the current form the conclusions about the domain should be lowered.

Relation between J and CNB domains

Results line 278-279: The 7.5 Å C1 structure shows that the J and CNB domains are close by each other. However the MD simulations presented do not seem to indicate that the J domain moves in a correlated way with its adjacent CNB-B domain. I do not see how panels 2B,F, and G show such a correlation.

Asymmetry between protomers

Results Line 314-315: The GaMD simulations seem to show less asymmetry in the wt holoenzyme and more in the RIIβ2J-C2 construct. This seems to be the opposite from the other MD simulations in Figure 2, A-E and Figure 3E and F, as well as Supplemental Figure 7. However here the authors state that the wt protomers are tightly coupled, the exact opposite of what had been said previously. These differing results should be discussed.

Figures

Figure 1: The choice of white as the map density makes it quite hard to see, particularly since the C domain model is also white. If possible it would be good to add some more contrast. It is particularly hard to see how well the density fits the J/A helix in panels D and E.

Supplementary Figure 12: The figure legend describes A,B, and C, but only A and B are shown, which correspond to B and C in the text. A, described in the text, seems to have been replaced by Supplementary Figure 10

Other

Line 142: Seems to be a word missing after “crystal” (symmetry perhaps?)

Lines 155-157: Mentioning of missing 14 residues is redundant (already explained in prior paragraph)

Line 314: “consistence” should be “consistent”

Line 472: no all-caps on “every”.

Recommendation

I recommend publication upon revision of the text. I do not recommend any new data be collected, provided the conclusions about the D/D location are weakened. The current conclusions would require stronger evidence in the cryo-EM map. The methodology seems clearly presented in the Supplemental methods, and the Supplementary Figures all support the paper and are presented well. The data have been submitted to the appropriate repositories.

---

## [Decision Letter · Decision Letter 2]

5 Nov 2020

Dear Susan,

Thank you for submitting your revised Research Article entitled "Capturing allostery: asymmetry and dynamics of PKA RIIβ holoenzyme with DnaJB1-PKAc fusion in fibrolamellar hepatoceullar carcinoma" for publication in PLOS Biology. I have now obtained advice from the four original reviewers and have discussed their comments with the Academic Editor.

We're delighted to let you know that we're now editorially satisfied with your manuscript. Nevertheless, we would like you to consider the following changes in the title for improvement: "Structural analyses of the PKA RIIβ holoenzyme containing the oncogenic DnaJB1-PKAc fusion protein reveal protomer asymmetry and fusion-induced allosteric perturbations in fibrolamellar hepatoceullar carcinoma."

In addition, we think that the abstract is a bit too specialised and we would like you to try to place the findings in a broader context to draw out the major new insights of your work in clearer manner, stepping away from too much detail. We have suggested some edits below as a working draft:

When the J-domain of the heat shock protein DnaJB1 is fused to the catalytic (C) subunit of cAMP-dependent protein kinase (PKA), replacing exon 1, this fusion protein, J-C-subunit (J-C), becomes the driver of Fibrolamellar hepatoceullar carcinoma. Here we use cryo-EM to characterize J-C bound to RIIβ, the major PKA regulatory (R) subunit in liver, thus reporting the first cryo-EM structure of any PKA holoenzyme. We report several differences in both structure and dynamics that could not be captured by the conventional crystallography approaches used to obtain prior structures. Most striking is the asymmetry caused by the absence of the second cyclic nucleotide binding domain and the J-domain in one of the RIIβ:J-C protomers. Using molecular dynamics simulations, we discovered that this asymmetry is already present in wt RIIβ 2C2 and had been masked in the crystal structure [WHAT FUNCTIONAL IMPLICATIONS DOES THIS HAVE/COULD HAVE? WHY IS THIS IMPORTANT?]. The cryo-EM structure, combined with SAXS, also allowed us to predict the general position the Dimerization/Docking (D/D) domain, which is essential for localization and interacting with membrane-anchored A-Kinase-Anchoring-Proteins (AKAPs). This position provides a multi-valent mechanism for interaction of the RIIβ holoenzyme with membranes and would be perturbed in the oncogenic fusion protein. The J-domain also alters several biochemical properties of the RIIβ holoenzyme: it is easier to activate with cAMP and the cooperativity is reduced. These results provide new insights into how the finely-tuned allosteric PKA-signaling network is disrupted by the oncogenic J-C-subunit, ultimately leading to the development of Fibrolamellar hepatoceullar carcinoma.

Before we can formally accept your paper and consider it "in press", we also need to ensure that your article conforms to our guidelines. A member of our team will be in touch shortly with a set of requests. As we can't proceed until these requirements are met, your swift response will help prevent delays to publication. Please also make sure to address the data and other policy-related requests noted at the end of this email.

- a cover letter that should detail your responses to any editorial requests, if applicable

*Copyediting*

*Published Peer Review History*

*Early Version*

Sincerely,

Ines

--

Ines Alvarez-Garcia, PhD

Senior Editor,

ialvarez-garcia@plos.org,

PLOS Biology

Fig. 2A, B; Fig. 3A, B; Fig. 4A-D; Fig. 5D, F; Fig. S2D; Fig. S5B-D; Fig. S7A-F and Fig. S11A, B 

Please make sure that the structural data you have deposited in the EMDB-PDB database is publicly available before the manuscript is accepted for production.

Reviewers’ comments

Rev. 1:

The authors have addressed my concerns and I support publication.

Rev. 2:

The authors have done a good job responding to the reviewers' comments. It is exciting to see the first cryo-EM PKA structure - I congratulate them on their work.

Rev. 3:

All my major concerns have been addressed. The changing of the text with regard to the biochemistry and analysis of the truncated subunit are now I believe clearer and more informative. This is an interesting piece of work and its great to see increased understanding of these large complex multi subunit structures. I hope the other reviewers also agree to accept.

Rev. 4: William Rice

The changes to the manuscript make it more clear and acceptable for publication as is.

---

## [Editor Report · Decision Letter 3]

18 Dec 2020

Dear Dr. Taylor,

I am writing concerning your manuscript submitted to PLOS Biology, entitled “Structural analyses of the PKA RIIβ holoenzyme containing the oncogenic DnaJB1-PKAc fusion protein reveal protomer asymmetry and fusion-induced allosteric perturbations in fibrolamellar hepatocellular carcinoma.”

We have now completed our final technical checks and have approved your submission for publication. You will shortly receive a letter of formal acceptance from the editor.

Kind regards,

PLOS Biology